# Integrin beta 1 facilitates non-enveloped hepatitis E virus cell entry through the recycling endosome

Rebecca Fu[1], Paula Jordan[1,10], Zoe Engels[1,10], Jasmin Alara Weihs [1], Josias Mürle[1], Huanting Chi [2], Sebastian Burbano de Lara [3], Barbara Helm[3], Mara Klöhn[4], Jungen Hu[1], Andrew Freistaedter[1], Tobias Boettler [5], Marco Binder [6], Ursula Klingmüller[3], Eike Steinmann[4], Pierre-Yves Lozach [7], Thibault Tubiana [8], Stanley M. Lemon [9] & Viet Loan Dao Thi [1] ✉

Hepatitis E virus (HEV) is a major cause of acute hepatitis and mainly transmitted faecal-orally. HEV particles present in faeces are naked (nHEV), whereas those found in the blood are quasi-enveloped (eHEV) with a cell-derived lipid membrane. Despite its global health impact, the cellular life cycle of HEV remains poorly understood, particularly regarding the mechanisms of viral entry into host cells. To address this knowledge gap, we develop a high content RNA-FISH-based imaging assay that allows for the investigation of the entry pathways of both naked and quasi-enveloped HEV particles. Surprisingly, we find that integrin α3, previously implicated in nHEV cell entry, is not expressed in the cell types that are most permissive for HEV infection. Instead, we identify integrin β1 (ITGB1) pairing with different α-integrins as the key player mediating nHEV cell entry. Our results indicate that the interaction of nHEV with ITGB1 facilitates entry through Rab11-positive recycling endosomes. In contrast, eHEV particles do not interact with ITGB1 and enter cells using a classical endocytic route via Rab5a-positive early endosomes. The entry of both types of HEV particles requires endosomal acidification and proteolytic cleavage by lysosomal cathepsins, which ultimately results in delivery of the HEV genome to the cytoplasm.

Hepatitis E virus (HEV) is a major cause of acute hepatitis[1]. HEV infections are usually self-limiting in healthy individuals but can become chronic in immunocompromised patients and cause a high mortality rate in pregnant women and patients with previous liver injury[1]. HEV has a single-stranded, positive RNA genome of 7.2 kb, encoding three open reading frames (ORF1, 2, and 3) (reviewed in[2]). ORF1 encodes the non-structural proteins responsible for viral replication, ORF2 the capsid protein, and ORF3 a small phosphoprotein that mediates

[1]Schaller Research Group, Department of Infectious Diseases, Virology, Heidelberg University, Medical Faculty Heidelberg, Heidelberg, Germany. [2]German Centre for Infection Research (DZIF), Partner Site Heidelberg, Heidelberg, Germany. [3]Division of Systems Biology of Signal Transduction (B200), German Cancer Research Center (DKFZ), Heidelberg, Germany. [4]Department of Molecular and Medical Virology, Ruhr University Bochum, Bochum, Germany. [5]Department of Medicine II, Medical Center – University of Freiburg, Breisgau, Germany. [6]Research Group "Dynamics of Early Viral Infection and the Innate Antiviral Response", Division Virus-Associated Carcinogenesis (D430), German Cancer Research Center (DKFZ), Heidelberg, Germany. [7]Univ. Lyon, INRAE, EPHE, IVPC, Lyon, France. [8]Institute for Integrative Biology of the Cell (I2BC), Université Paris-Saclay, CEA, CNRS, Gif-sur-Yvette, France. [9]Departments of Medicine and Microbiology & Immunology, The University of North Carolina at Chapel Hill, Chapel Hill, NC, USA. [10]These authors contributed equally: Paula Jordan, Zoe Engels. ✉e-mail: VietLoan.DaoThi@med.uni-heidelberg.de

secretion of viral progeny. There are eight HEV genotypes (HEV-1 to −8), of which five (HEV-1 to −4 and HEV-7) are capable of infecting humans[3]. Of note, HEV-3, which is the predominant genotype in developed countries, can infect a broad range of animals, including pigs, deer, and rabbits, and can be transmitted zoonotically to humans. In vitro, HEV-3 can infect a wide range of non-hepatic cell types (reviewed in[4]).

HEV is usually transmitted faecal-orally through contaminated food and water[1]. It enters and leaves the human body in its naked form (nHEV) and circulates in the blood wrapped in a host-derived lipid quasi-envelope (eHEV) acquired during virus budding from cells (reviewed in[5]). eHEV particles in the blood are protected from neutralising antibodies, while nHEV particles shed in the faeces facilitate transmission to another host[6]. This directional release is mediated by the polarisation of hepatocytes in vivo, from which apically co-secreted bile acids strip off the envelope as the progeny particles are released into the bile[7]. The quasi-envelope confers the particles a lower buoyant density, making them approximately ten times less infectious than the naked virions[7–9]. In this respect, HEV is very similar to hepatitis A virus (HAV), which is also enterically transmitted and exists in both, naked and quasi-enveloped forms[10].

Despite growing interest, fundamental steps of the HEV life cycle, including the cell entry pathways of both nHEV and eHEV particles are poorly understood. Several host factors, such as heparan sulfate proteoglycans, glucose-regulated protein 78, and asialoglycoprotein receptor have been proposed as nHEV attachment factors (reviewed in[11]). While an HEV entry receptor has not yet been identified, epidermal growth factor receptor (EGFR) has been shown to modulate HEV entry in human hepatocytes[12]. Additionally, a recent study showed that the T cell immunoglobulin mucin domain-1 (TIM-1) receptor promotes infection of eHEV particles through binding to phosphatidylserines which are present on their quasi-envelopes[13].

Shiota and colleagues have previously demonstrated a correlation between integrin α3 (ITGA3) expression and nHEV permissiveness in the PLC/PRF/5 hepatoma cell line as well as a direct interaction between nHEV particles and the ITGA3 ectodomain[14]. Integrins are cell-matrix adhesion receptors that always assemble and function as αβ heterodimers. ITGA3 only pairs with integrin β1 (ITGB1). In contrast, ITGB1 is the most promiscuous integrin subunit and is expressed in a range of different cell types and tissues, where it can form heterodimers with integrins α1 to α11 and αv (reviewed in[15]). As multifunctional cell surface molecules, integrins have been shown to act as entry receptors for a variety of viruses, allowing binding to the cell and thereby stimulating viral endocytosis (reviewed in[16]). For example, both naked and quasi-enveloped HAV particles undergo clathrin-dependent endocytosis facilitated by ITGB1[17], and binding of the adenovirus penton base capsid protein to integrins initiates virus internalisation by stimulating endocytosis via clathrin-coated vesicles[18]. Following internalisation, the majority of endocytosed integrins are recycled back to the cell surface, while a smaller fraction is destined for degradation in the lysosome[19].

Studies on the involvement of endosomal trafficking of HEV particles have been contradictory. Following potential receptor interactions, Yin and colleagues found that both eHEV and nHEV particles are internalised via clathrin-dependent endocytosis[8]. The authors also proposed that eHEV entry requires the small GTPases Rab5 and Rab7, typical markers of early and late endosomes, respectively, whereas nHEV entry does not appear to rely on the endocytic pathway. In contrast, a study by Holla *et al.* showed that naked hepatitis E virus-like particles (HEV-LPs) are trafficked to Rab5-positive compartments and degraded in lysosomal compartments[20].

An in-depth understanding of the molecular steps of HEV cell entry is missing, partially due to the lack of suitable assays. Unlike enveloped viruses, HEV particles lack glycoproteins and cannot be studied using viral pseudotypes. The icosahedral capsid is compact and fusion with fluorescent tags is likely to affect both progeny assembly and receptor interactions during entry.

Here, in this work, to address this knowledge gap and the controversies surrounding HEV cell entry, we develop a high-content imaging approach to visualise and study the entry pathways of both nHEV and eHEV particles. Our study clarifies and extends previous reports, and further reveals a critical role for the cell surface receptor ITGB1 in pairing with an exchangeable α-integrin partner to determine the endocytic trafficking pathways of nHEV, but not eHEV particles.

## Results

### ITGB1 heterodimerises with α-integrins in a cell type-dependent manner and mediates nHEV but not eHEV cell entry

ITGA3 was reported previously to be an essential host factor for nHEV particle entry into PLC/PRF/5 hepatoma cells[14]. We infected ten different liver cancer cell lines, for which we have performed complete proteomic analyses (Fig. 1A), with density-gradient purified nHEV particles (1.25 g/cm³, Suppl. Figure 1). Since the most established and accurate practice for quantifying viral infection events is to report plaque- or focus-forming units (FFU)[21], we aimed to quantify and compare nHEV FFUs upon infection of the different liver cell lines. To reliably assess the number of FFUs, cells were inoculated with a highly diluted viral suspension (MOI = 0.1 GE/cell) to obtain well-separated and countable foci (Suppl. Figure 2).

In agreement with Shiota et al., we found that ITGA3 was expressed in PLC/PRF/5 cells. However, we did not observe ITGA3 expression in either HepG2 or Huh-1 cells, despite these being the most nHEV-permissive cell lines tested. To confirm these results, we performed Western blot analysis and investigated selected integrin expression in hepatoma cell lines which are commonly used for HEV research, including Huh-7, the Huh-7 subclone S10-3[22], and the HepG2 subclone HepG2/C3A[23]. As shown in Fig. 1B, we detected little endogenous ITGA3 expression in either primary human hepatocytes (PHH) or any of the cell lines tested. Ectopic ITGA3 expression in S10-3 cells allowed us to confirm the suitability of the anti-ITGA3 antibody used for this analysis (Suppl. Figure 3A). Instead, we found ITGB1 as well as integrins α2 (ITGA2), α5 (ITGA5), and α6 (ITGA6) to be highly expressed. Furthermore, high-resolution confocal microscopy analysis revealed that the majority of ITGA3 was localised intracellularly (Suppl. Figure 3B). In contrast, ITGB1, ITGA2, and ITGA5 were mainly expressed at the cellular plasma membrane where they often colocalised (Suppl. Figure 3B). Altogether, these results suggested that ITGA3 is not essential for mediating nHEV entry into Huh-7 based cell lines.

To further corroborate our results, we used CRISPR-Cas9 to knock out ITGA2 in Huh-7 cells (Fig. 1C). As shown in Fig. 1D (non-normalised data in Suppl. Figure 4), ITGA2 knockout (KO) cells were substantially less permissive than the parental cells, suggesting that other α-integrins besides ITGA3 can be involved in nHEV infection. To corroborate these results, we used a specific ITGA2 inhibitor and found that it inhibited nHEV infection in a dose-dependent manner, albeit with low efficacy (Fig. 1E, non-normalised data in Suppl. Figure 4).

Since ITGB1 mediates entry of many viruses[24–26] and is known to form functional heterodimers with different α-integrins, including ITGA2 and ITGA3, we further investigated whether ITGB1 itself plays a role in HEV infection. Using CRISPR-Cas9, we generated ITGB1 KO cells, selected two clones (Fig. 1F), and infected them with nHEV.

In conventional non-polarised HEV-replicating cell culture systems, naked HEV particles can be recovered from cell lysates, whereas quasi-enveloped particles are released into the supernatant. Since the quasi-envelope of eHEV particles interferes with antibody recognition[7] and therefore probably also with capsid-receptor interactions, we challenged these cells in parallel with density-gradient purified eHEV particles (1.12 g/cm³, Suppl. Figure 1). Throughout this study, we applied a higher number of genome equivalents per cell (GE/cell) for eHEV compared to nHEV to achieve comparable FFUs. This adjustment

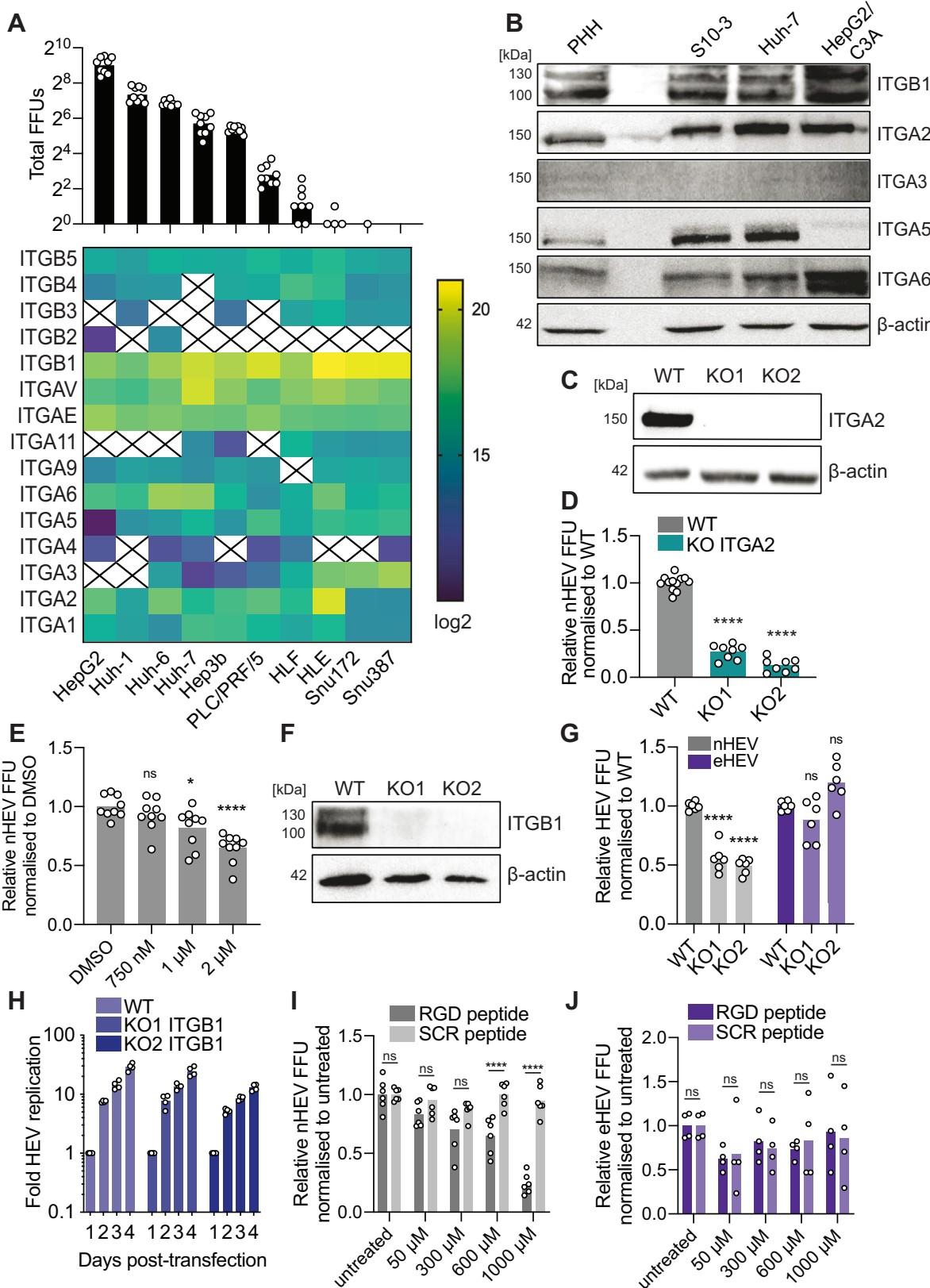

was necessary due to the approximately 100-fold higher specific infectivity of nHEV compared to eHEV particles (Suppl. Figure 5), consistent with our previous findings[7]. To account for this difference, we used 50 times more eHEV particles per cell (5 GE/cell for eHEV vs. 0.1 GE/cell for nHEV), allowing us to approximate FFU counts. Due to the volume limitations of eHEV particles that can be added per well, it

was not possible to achieve a corresponding 100-fold increase in our infection assays.

nHEV infection of ITGB1 KO cells was significantly reduced compared to wild-type (WT) cells, as assessed by counting HEV FFUs 5 days post-infection (Fig. 1G, non-normalised data in Suppl. Figure 4). By contrast, eHEV infection appeared to be unaffected by the absence of

**Fig. 1 | ITGB1 heterodimerises with α-integrins in a cell type-dependent manner and mediates nHEV but not eHEV cell.** (A) Heat map showing log2-transformed expression levels of selected integrins in uninfected liver cell lines determined by mass-spectrometry-based proteomics (lower panel, n = 3). Protein levels not detected in at least two replicates were excluded (represented by an X). Cell lines were infected with density gradient-purified nHEV (MOI = 0.1 GE/cell) and infectivity was assessed by staining against HEV ORF2 protein and quantifying FFUs 5 days post-infection (upper panel). n = 9 replicates. (B) Western blot (WB) analysis of hepatoma cell lines and PHH lysates. (C) WB analysis of Huh-7 ITGA2 WT and KO cell lysates. (D) Huh-7 ITGA2 WT and KO cells were infected with nHEV (MOI = 0.1 GE/cell). Infectivity was assessed as in (A). n = 8 replicates. (E) Huh-7 cells were infected with nHEV (MOI = 0.1 GE/cell) in the presence of an integrin α2β1 inhibitor BTT3033 or DMSO. Infectivity was assessed as in (A). n = 9 replicates. (F) WB analysis of S10-3 ITGB1 WT and KO cell lysates. (G) S10-3 ITGB1 WT and KO cells were infected with nHEV (MOI = 0.1 GE/cell) or eHEV particles (MOI = 5 GE/cell). Infectivity was assessed as in (A). n = 6 replicates. (H) S10-3 ITGB1 WT and KO cells were electroporated with an HEV-Gaussia luciferase (GLuc) replicon and HEV replication was quantified by measuring GLuc activity in the supernatant during days 1 to 4 post-electroporation. n = 4 replicates. (I) and (J) S10-3 WT cells were infected with nHEV (MOI = 0.1 GE/cell, n = 6) or eHEV particles (MOI = 5 GE/cell, n = 4) in the presence of an RGD-containing peptide or a scrambled (SCR) control peptide. Infectivity was assessed as in (A). All replicates are from two (B, H, J) or three (A, C, D, E, F, G, I) independent experiments. Statistical analysis was performed by one-way (D, E, G) or two-way ANOVA (I, J). *: $p < 0.05$; ****: $p < 0.0001$; ns, non-significant.

ITGB1. We next electroporated ITGB1 KO and WT cells with a sub-genomic Gaussia luciferase (GLuc) reporter HEV replicon and quantified secreted GLuc as a surrogate for HEV replication. Compared to WT cells, HEV replication was not significantly impaired in ITGB1 KO cells (Fig. 1H), consistent with a role for ITGB1 in HEV cell entry.

To confirm this universal role of ITGB1 in nHEV entry, we transfected other hepatoma and non-hepatoma cell lines, such as Huh-7, HepG2/C3A, lung A549, and intestinal Caco-2 cells with small interfering RNA (siRNA) targeting ITGB1 (siITGB1) or non-targeting siRNA (siNT) 48 h prior to HEV inoculation. In line with our observations in S10-3 cells, we found that knockdown of ITGB1 significantly reduced nHEV, but not eHEV infection, of all cell lines tested (Suppl. Figure 6A–C).

ITGB1 heterodimers, depending on the α-integrin partner, act as laminin, leukocyte, and collagen receptors[27]. Integrin α2β1 is a collagen receptor which is known to bind to ligands through their cell adhesion arginine-glycine-aspartic acid (RGD) motif[28,29]. Since we observed a reduction in nHEV infection upon depletion of α2 and β1 integrin subunits, we asked whether entry could be blocked by a competing RGD peptide. WT cells were treated with either an RGD-containing peptide GRGDNP or control scrambled peptide GRADSP for 15 min prior to inoculation with n/eHEV in the presence of the peptides for 6 h. Application of the RGD, but not the control peptide, during entry resulted in a dose-dependent inhibition of nHEV (Fig. 1I, non-normalised data in Suppl. Figure 4) but not eHEV infection (Fig. 1J, non-normalised data in Suppl. Figure 4). We also confirmed the inhibitory effect of the RGD peptide on nHEV entry in Huh-7 and HepG2/C3A cells (Suppl. Figure 6E). Since the laminin receptor integrin α3β1 generally binds to ligands in an RGD-independent manner[30,31], our data further corroborate the notion that ITGA3 is not critical for nHEV entry into these cell lines.

Collectively, our results demonstrate that ITGB1 is a key player in mediating nHEV but not eHEV entry, while the α-integrin partner required to form a functional integrin heterodimer appears to be exchangeable in a cell line-specific manner.

## RNA-fluorescence in situ hybridization allows detection of single HEV particles

Previous studies of HEV cell entry have relied on either the use of HEV-LPs[20] or the quantification of infected cells[8]. However, HEV-LPs are derived from self-assembled truncated recombinant ORF2 proteins (reviewed in[32]) and lack the C-terminus of the critical protruding "P" domain, which could modulate virus binding to cell receptors[33]. In addition, the study of cell entry requires separation from the other steps of the viral life cycle, such as genome replication. In the absence of a traditional lipid envelope containing viral glycoproteins and with the aim of studying the role of ITGB1 during nHEV entry, we developed a high-content imaging assay to study incoming HEV particles based on the detection of individual HEV genomes by RNA-fluorescence in situ hybridisation (RNA-FISH) (Fig. 2A).

First, we allowed density-gradient purified nHEV and eHEV particles to bind to hepatoma S10-3 cells on ice. Since it was previously reported that eHEV particles have a > 10-fold lower cell-binding efficiency at 4 °C than nHEV particles[8], we allowed nHEV and eHEV particles to bind for 2 h and 6 h, respectively. Similar to the FFU assays, we could not apply the same MOI for nHEV and eHEV for these assays since the concentration of eHEV particles cannot be increased to the same extent as for nHEV particles due to the high volume and low specific infectivity of eHEV particles harvested from cell culture supernatants.

The cells were then either fixed to analyse "binding" or rapidly heated to 37 °C to induce virus "internalisation" and fixed 6 h later. After fixation, we stained and detected individual nHEV and eHEV genomes by RNA-FISH either located at the cell periphery (Fig. 2B, upper image row) or translocated into the cell interior (Fig. 2B, lower image row). As a membrane marker to visualise the cell periphery, we co-stained ITGB1 (magenta). We then calculated the amount of HEV particles per cell by dividing the number of HEV genomes detected by the number of nuclei in an image frame for both conditions (Fig. 2B, right panel). In order to ensure the specificity of the assay, we used the ORF1 probe for hybridisation of mock-infected cells, which gave no detectable signal (Suppl. Figure 7A). In addition, we used a control GAPDH probe, which did not detect ex vivo nHEV particles (Suppl. Figure 7B), but detected genomic RNA in S10-3 cells (Suppl. Figure 7C).

To ensure that the HEV genomes detected were in capsid-formed particles, we pre-treated nHEV particles with convalescent HEV patient serum containing anti-ORF2 antibodies[34] or with non-HEV patient serum as a control. While the control serum had little effect, we observed an almost 100% reduction in detected nHEV genomes after neutralisation with the patient serum (Fig. 2C). In addition, we pre-treated nHEV particles with RNAse A prior to inoculation into S10-3 cells. As shown in Suppl. Figure 7D, the RNAse treatment did not reduce the amount of HEV genomes, thus excluding the detection of free HEV genomes in our assay. We further treated the cells with the replication inhibitor NITD008[35] and found no difference in the number of HEV RNA particles as compared to the DMSO control (Suppl. Figure 7E), demonstrating that we did not detect newly synthesised RNA or progeny virus.

Finally, to ensure that viral particles packaged full-length HEV genomes, rather than defective, truncated, or subgenomic genomes, we used RNA-FISH probes targeting the full-length HEV genome spanning both ORF1 (green probe) and ORF2 (magenta probe) (Fig. 2A). As shown in Fig. 2D, multiplexing of both probes revealed that the majority of detected particles contained the full-length HEV genomes, while only a fraction of particles contained either truncated (ORF1 only) or potentially subgenomic (ORF2 only) genomes. These results show that RNA-FISH allows the detection of individual viral particles and can be used to study the cell entry routes of infectious HEV particles.

## ITGB1 interacts with a DGR motif in the protruding capsid domain leading to the internalisation of nHEV particles

We next sought to validate the role of integrins in HEV entry by studying incoming HEV particles by RNA-FISH (Fig. 3). Since the α-

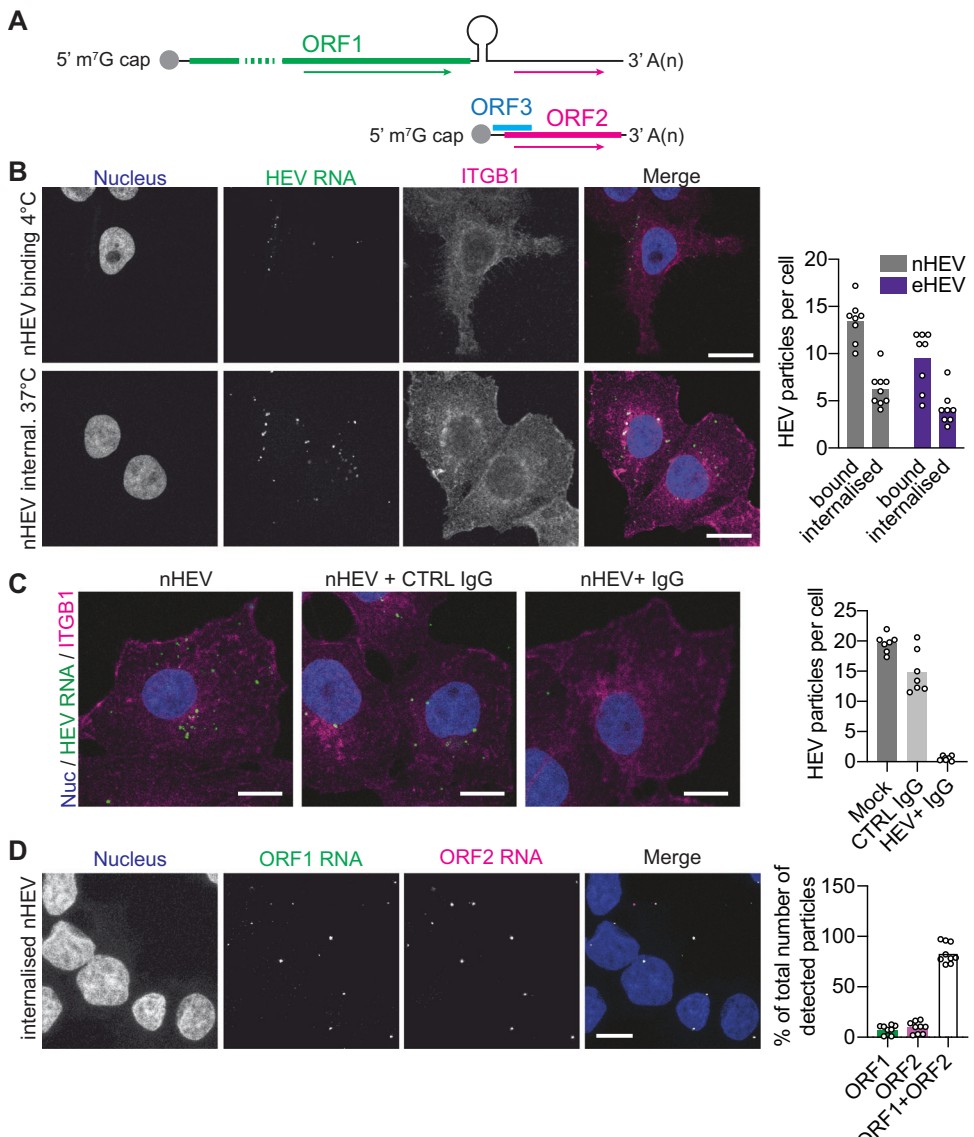

**Fig. 2 | RNA-fluorescence in situ hybridization allows the detection of single HEV particles.** (**A**) Scheme of HEV genome organization showing RNA-fluorescence in situ hybridization (RNA-FISH) probes targeting either the full-length genome (green arrow) or the full-length and subgenomic genome (pink arrow). (**B**) Prechilled hepatoma cells S10-3 were inoculated with nHEV (MOI = 30 GE/cell) or eHEV particles (MOI = 20 GE/cell) and incubated for 2 h and 6 h, respectively, on ice to allow particle binding. The cells were then either fixed or the inoculum removed and shifted to incubation at 37 °C for 6 h to allow particle internalisation. After binding or internalisation, the cells were fixed and stained with DAPI (blue, nucleus) and against ITGB1 (magenta); HEV genomes (green) were detected by RNA-FISH using the ORF1 probe. n = 8 microscope fields (-90 cells). Scale bar = 20 μm. (**C**) nHEV particles were pre-incubated with convalescent HEV patient serum (1:1000) containing anti-ORF2 antibodies or with non-HEV patient serum as control for 1 h at 37 °C. S10-3 cells were inoculated with mock or serum-treated nHEV (MOI = 30 GE/cell) at 37 °C for 6 h. Cells were then fixed and stained as described in (**B**). n = 7 microscope fields (-90 cells). Scale bar = 10 μm. (**D**) S10-3 cells were inoculated with nHEV (MOI = 10 GE/cell) in incubated for 6 h at 37 °C followed by fixation. HEV genomes were detected by RNA-FISH using both ORF1 (green) and ORF2 (magenta) probes. n = 9 microscope fields (-90 cells). Scale bar = 10 μm. All images represent single slices of confocal images. The detected HEV genomes were quantified using CellProfiler. HEV particles per cell were calculated by dividing the total number of detected HEV genomes by the number of nuclei in an image frame. All data are from three independent experiments.

integrin partner appeared to be exchangeable, we focused on ITGB1. We inoculated WT and ITGB1 KO cells with nHEV and eHEV particles on ice, followed by inoculum removal and internalisation at 37 °C. In agreement with results shown in Fig. 1G–J, the number of bound and internalised nHEV particles (Fig. 3A) but not eHEV particles (Fig. 3B) was significantly lower in ITGB1 KO cells than in WT cells. To confirm that the reduction in nHEV entry in ITGB1 KO cells was specifically due to the absence of ITGB1, we ectopically rescued the expression of ITGB1 (Suppl. Fig. 8A). In these cells, nHEV binding and internalisation was restored similar to WT levels (Fig. 3A).

Next, we transfected S10-3 cells with siRNA targeting ITGB1 (siITGB1) or siNT 48 h prior to HEV inoculation (Suppl. Fig. 8B). Consistent with our results with the ITGB1 KO cells, we found that the knockdown significantly reduced nHEV, but not eHEV, binding and internalisation (Suppl. Fig. 8C). We also used siRNA to downregulate the focal adhesion kinase (FAK). All β1-integrins are capable of activating FAK, which leads to the recruitment of additional cellular factors and results in rearrangements of the cell cortex and cytoskeleton[36]. As shown in Fig. 3D, knockdown of FAK by siRNA (Fig. 3C) significantly reduced nHEV but not eHEV internalisation.

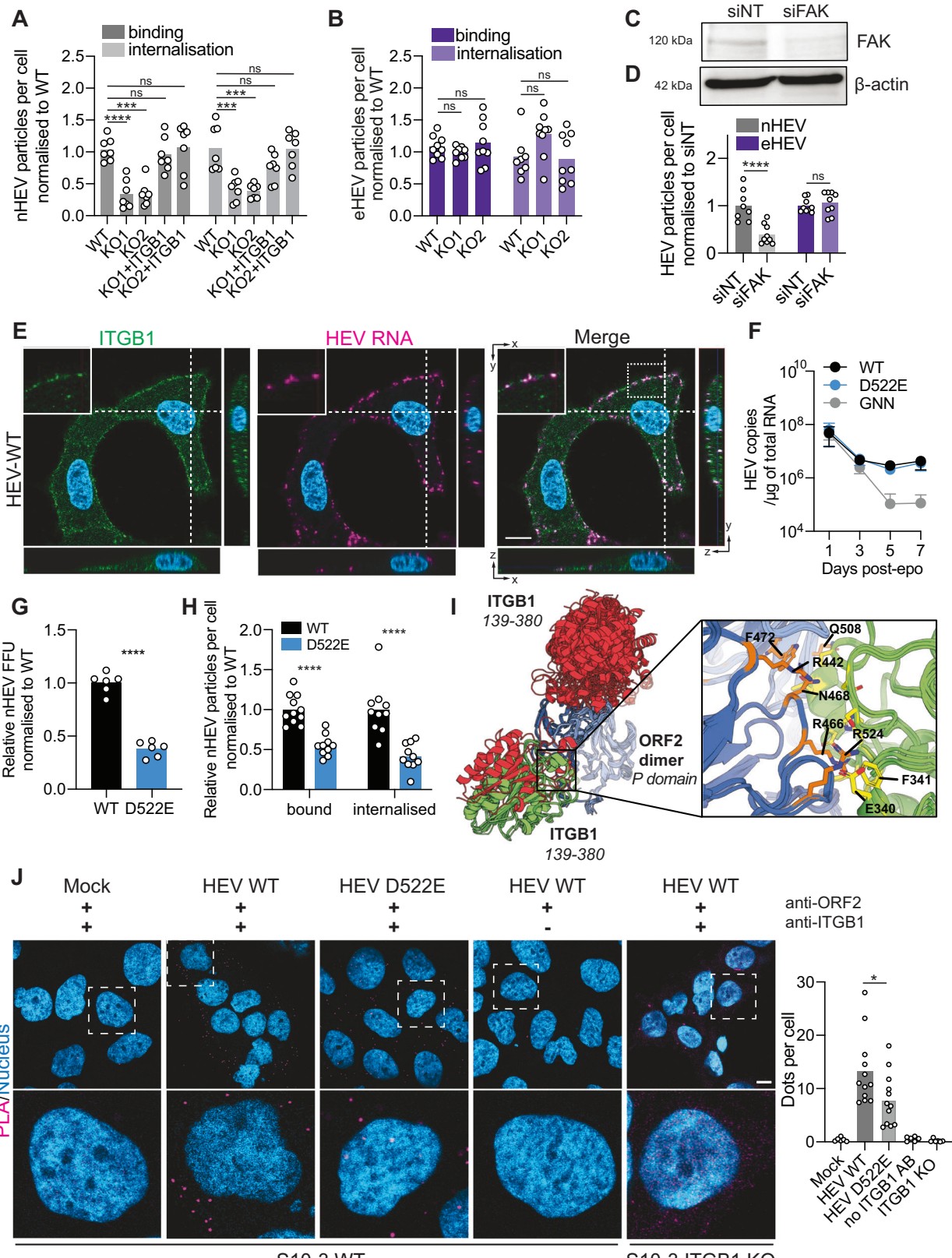

To further investigate a potential interaction of the HEV capsid with ITGB1, we detected colocalisation of nHEV particles, but not eHEV particles, with ITGB1 on the cell membrane (Fig. 3E and Suppl. Fig. 9A, B). To further corroborate this potential interaction, we identified a well-exposed reverse RGD "DGR" motif in the protruding P-domain of the ORF2 sequence (Suppl. Fig. 10A), which is highly

conserved across the *Orthohepeviridae A*. DGR sequences have poor binding coefficients for the RGD integrin ligand site. However, the aspartic acid can undergo spontaneous nonenzymatic deamination to isoaspartic acid (*iso*D) and the resulting IsoDGR sequence rivals that of "RGD" containing peptides[37]. It has been shown that *iso*D occurs naturally in ECM proteins, including β1-integrins ligands collagen and

**Fig. 3 | ITGB1 interacts with a DGR motif in the protruding capsid domain leading to the internalisation of nHEV particles.** (**A**) and (**B**) S10-3 ITGB1 WT and KO cells ± ectopic ITGB1 were inoculated with (**A**) nHEV (MOI = 30 GE/cell) for (**B**) 2 h or eHEV (MOI = 20 GE/cell) for 6 h on ice (binding). After inoculum removal, cells were shifted to 37 °C for (**A**) 6 h or (**B**) 24 h (internalisation). HEV genomes were detected by RNA-FISH (version 2 kit with ORF1 probe) and quantified using CellProfiler. n = 7 (**A**) and n = 9 (**B**) microscope fields (-120 cells). (**C**) and (**D**) S10-3 cells were transfected with siRNAs against FAK or a non-target control and 48 h later, (**C**) harvested for WB or (**D**) inoculated with nHEV (MOI = 30 GE/cell) or eHEV (MOI = 20 GE/cell). After 8 h at 37 °C, cells were analysed as described in (**A**). n = 8 microscope fields (-80 cells). (**E**) Maximum projections of S10-3 cells inoculated with nHEV (MOI = 20 GE/cell) for 2 h on ice. HEV particles (magenta) were visualised by RNA-FISH as in (**A**), followed by ITGB1 staining (green) and DAPI (blue). (**F**) Genomes of WT, D522E, and replication incompetent GNN HEV were quantified in electroporated S10-3 cells over time. n = 4 replicates. (**G**) S10-3 cells were infected with WT or D522E nHEV (MOI = 0.1 GE/cell) and analysed by FFU staining at 5 days post-infection. n = 6 replicates. (**H**) S10-3 cells were inoculated with WT or D522E nHEV (MOI = 30 GE/cell) for analysis of binding or internalisation as described in (**A**). n = 10 microscope fields (-80 cells). (**I**) Spatial arrangement of ITGB1$_{139-380}$ relative to the ORF2 P-domain dimer (blue). Green models represent interactions with low PAE$_{inter}$ values, red models indicate high PAE$_{inter}$ values (shown in Suppl. Fig. 12C). Close-up view (rendered with PyMol) shows the interface between green ITGB1$_{139-380}$ models and ORF2-P. (**J**) S10-3 ITGB1 WT or KO cells were inoculated with WT or D522E nHEV (MOI = 50 GE/cell) for 2 h on ice (binding) and processed for proximity ligation assay. n = 6 microscope fields (controls), n = 12 microscope fields (HEV WT and D522E) (-90 cells). Scale bar=10 µm. All data are from two (**F, G, J**) or three (**A, B, C, D, E, H**) independent experiments. Statistical analysis was performed by unpaired two-tailed Student's t test (**G**), one-way (**A, B, D, J**) or two-way ANOVA (**H**). *: p < 0.05; ***: p < 0.001; ****: p < 0.0001; ns, non-significant.

---

fibronectin[38,39]. Peptides containing the DGR sequence have an inhibitory effect on cell attachment to collagen-coated surfaces[40].

To study if the DGR motif likewise mediates interaction between the HEV capsid and ITGB1, we then mutated the aspartic acid to glutamic acid (D522E). We deliberately chose a mild mutation rather than a more disruptive change, such as mutating the aspartic acid to an alanine. First, we verified that the mutation did not impact genome replication (Fig. 3F). Of note, at least part of the basal HEV RNA detected up to day 3 post-electroporation resulted from high levels of incoming, electroporated RNA, as revealed by comparison with a replication-incompetent GNN mutant of the RNA-dependent RNA polymerase. We also density-gradient purified both WT-HEV and D522E-HEV particles (Suppl. Fig. 10B) and found that the mutation did not impact progeny assembly (Suppl. Fig. 10C). When applying equal amounts of genome equivalents, we found that D522E-nHEV was less infectious than WT-nHEV (Fig. 3G, non-normalised data in Suppl. Figure 4). In addition, D522E-nHEV particles were impaired in binding to cells and internalisation (Fig. 3H). The mutation did not reduce infectivity or entry of D522E-eHEV particles (Suppl. Fig. 10D, E), allowing us to rule out deleterious effects of the mutation on capsid assembly. We further found that D522E-nHEV was less sensitive to ITGB1 depletion and RGD peptide treatment than WT-nHEV (Suppl. Fig. 11). Altogether, these results strongly suggested an interaction between the HEV capsid via the DGR motif and ITGB1.

To further support this hypothesis, we modelled the interaction between ITGB1 and HEV ORF2 using Alphafold2. The HEV capsid is composed of capsomeres formed by homodimers of a single structural capsid protein, which are thought to protrude from the viral surface and interact with host cells to initiate infection[41]. Given that AlphaFold demonstrates improved performance in predicting interactions for smaller regions, we generated various models using both monomeric and dimeric forms of the ORF2 P-domain, alongside the head domain of ITGB1 (spanning residues 139-380). Our goal was to identify a high-scoring binding interface with low Predicted Aligned Error (PAE). The interaction with the monomeric ORF2-P form yielded high PAE scores (Suppl. Fig. 12A, B), whereas the interaction with the dimeric ORF2-P form resulted in low PAE scores (Suppl. Fig. 12C). We identified two primary binding modes at the interface of the ORF2-P dimer (Fig. 3I). The first mode positioned ITGB1$_{139-380}$ on the dimer with direct accessibility to the HEV capsid, but yielded high PAE values (Suppl. Fig. 12B). The second mode showed a strong predicted interaction supported by low PAE values (Suppl. Fig. 12C). Although this binding site to ORF2 appeared to be less accessible in the context of the fully assembled capsid (Suppl. Fig. 12D), a proline-rich hinge region between the P- and the M-domains identified in the HEV-LP structure[33] is likely to provide the necessary flexibility. The predicted interface between ITGB1$_{139-380}$ and ORF2-P displayed a predominantly positively charged surface enriched with arginine residues, along with some hydrophobic contributions (Fig. 3I). Based on this molecular

environment, six amino acids—Q508, F472, R442, N468, R466, and R524—were selected for alanine mutagenesis using AlphaFold2. These mutations were evaluated both individually and in combination to assess their impact on the quality of the models. Notably, R524 emerged as the single point mutation with the most significant impact on PAE scores (Suppl. Fig. 12E). Additional mutations alongside R524 further exacerbated this effect, underscoring its potential role in mediating the ORF2-P/ITGB1 interaction. Although more work is needed to prove the interaction between the HEV capsid and ITGB1 as modelled by Alphafold, these results support the DGR motif as a potential binding motif for ITGB1.

Finally, we confirmed the potential interaction between the HEV capsid and ITGB1 using a proximity ligation assay. As shown in Fig. 3J, we observed signals only when WT but not ITGB1 KO cells were inoculated with WT nHEV particles. This proximity was impaired when the DGR motif in the HEV capsid was mutated. Consistent with this finding, we also detected less colocalisation of D522E-nHEV than of WT-nHEV particles with ITGB1 (Suppl. Fig. 9A, B), further highlighting the potential role of this motif in nHEV cell entry.

## Co-detection of capsid and RNA allows analysis of HEV entry dynamics

Successful viral entry requires the release of the viral genome into the cytoplasm where the incoming genome can be translated into non-structural proteins to initiate genome replication. This process requires dissociation of the viral genome from the capsid. To study the kinetics of HEV uncoating, we combined viral genome detection by RNA-FISH with capsid staining using a specific anti-ORF2 antibody. First, we imaged cell-free nHEV particles and observed that the majority of HEV genomes overlapped with capsids, while only a fraction of capsids appeared to be devoid of HEV genomes (Fig. 4A).

We then imaged nHEV and eHEV particles after binding to cells (0 h) and 24 h after internalisation (Fig. 4B and Suppl. Fig. 13). Initially, we observed that all detected HEV genomes colocalised with the capsids, whereas 24 h later, most HEV genomes no longer colocalised with the capsids, indicating successful uncoating (Fig. 4B).

In order to confirm whether the genomes were delivered to the cytoplasm, we used a subcellular fractionation kit based on differential membrane permeabilisation to separate all cellular membranes, including endosomal membranes, from the cytosol in HEV-infected cells (Fig. 4C). At 24 h post-infection, we found that more than 50% of the total internalised HEV genome was released into the cytosol, compared to less than 20% after 3 h of infection (Fig. 4D).

Next, we investigated the kinetics of HEV capsid uncoating by quantifying HEV genomes and capsids over time post-internalisation (Fig. 4E, F). For nHEV, the number of capsids detected decreased steadily over the observation period. In agreement with our observations in Fig. 4B, we detected almost no capsids after 24 h anymore. In contrast, the number of HEV genomes remained relatively stable over

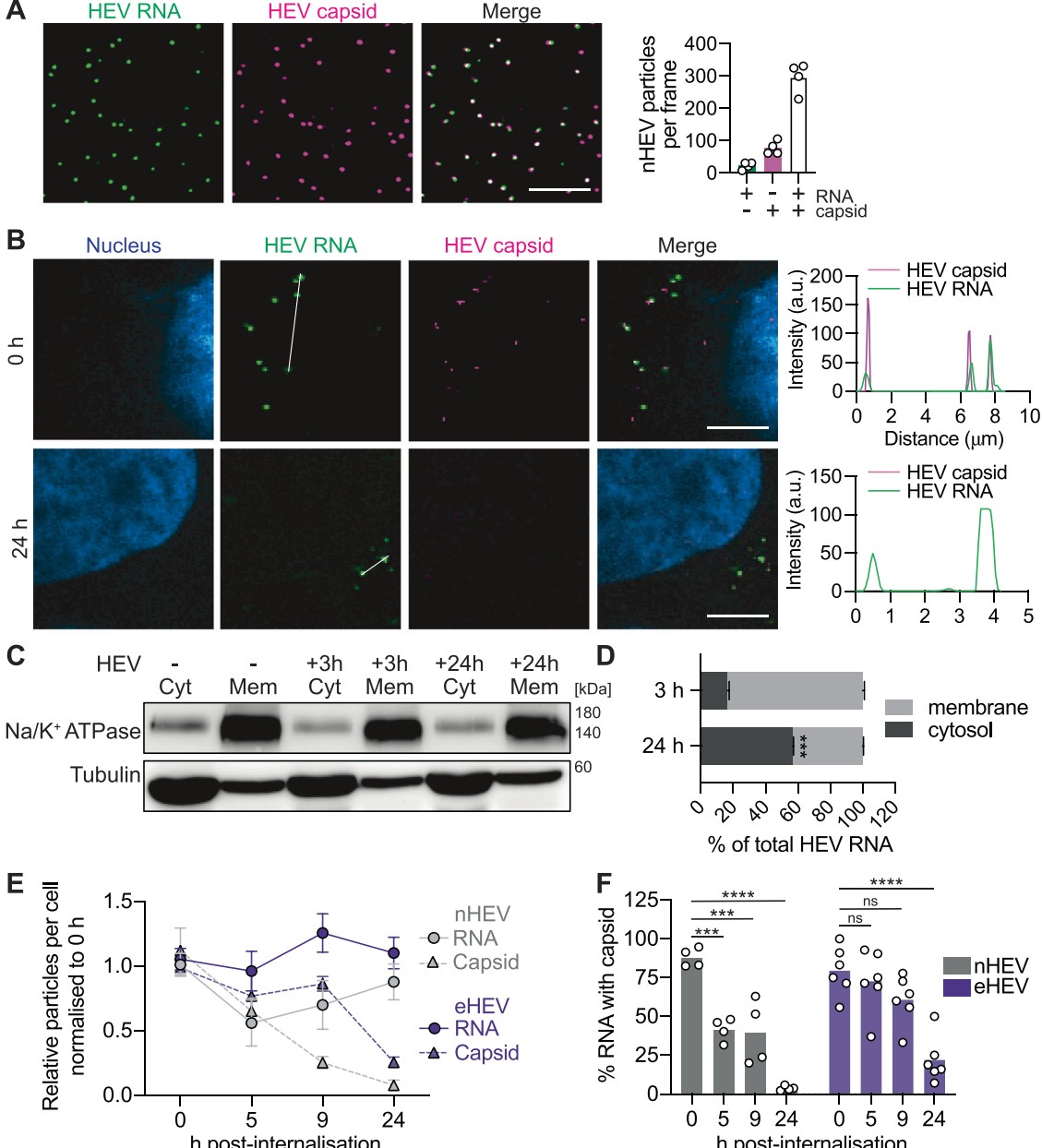

**Fig. 4 | Co-detection of HEV capsid and genome allows analysis of the dynamics of nHEV and eHEV entry.** (**A**) nHEV particles were immobilized on slides; capsids (magenta) were detected by immunofluorescence (IF) staining and HEV genomes (green) by RNA-FISH (version 1 kit with ORF1 probe). n = 4. Scale bar = 10 μm. (**B**) S10-3 cells were inoculated with nHEV (MOI = 30 GE/cell) and incubated for 2 h at 4 °C to allow binding (upper image row) followed by inoculum removal and 24 h at 37 °C for internalisation (lower image row). After fixation, cells were stained as described in (**A**). Line graphs on the right show the fluorescence intensities of capsids and genomes across the region of interest indicated by the white line in the images. Scale bar = 5 μm. (**C**) S10-3 cells were infected with nHEV (MOI = 50 GE/cell) and harvested after 3 h or 24 h followed by cell fractionation. Equal volumes from each fraction were analysed by WB against Na/K+ ATPase and tubulin. (**D**) Percentages of HEV genomes in each fraction in cells infected for 3 or 24 h. n = 6. (**E**)

S10-3 cells were inoculated with nHEV (MOI = 30 GE/cell) or eHEV (MOI = 20 GE/cell) for 2 h or 6 h, respectively, followed either by fixation (0 h) or internalisation at 37 °C and fixation at indicated time points. HEV capsids and genomes were detected as described above and quantified using CellProfiler. n = 4 microscope fields for capsid and n = 6 microscope fields for RNA. (**D**, **E**) Data presented as mean ± standard derivation (SD). (**F**) Calculated percentages of HEV genomes colocalising with capsids (from **E**) out of the total number of detected genomes per cell. n = 4 microscope fields for nHEV and n = 6 microscope fields for eHEV. Statistical analysis was performed by unpaired two-tailed Student's t test (**D**) or two-way ANOVA (**F**). ***: $p < 0.001$; ****: $p < 0.0001$; ns, non-significant. Images in (**A**) and (**B**) are maximum projections of 4 slices and 2 slices, respectively. All data are from two (**A**, **C**, **D**), three (**E**, **F**) or six (**B**) independent experiments.

the observation period. For eHEV, the number of capsids detected also decreased steadily. We have previously found that HEV replication initiates around 24 h post infection[42], which allowed us to rule out the possibility of newly synthesised genomes and/or capsids being detected in this observation window. Overall, our RNA-FISH-based assay allowed us to identify the approximate time window required for nHEV and eHEV particles to enter cells.

## Both eHEV and nHEV enter through the endocytic pathway but take differential routes

Upon ligand binding, integrins can be endocytosed in a clathrin-dependent manner and either degraded or recycled back to the plasma membrane[36]. We therefore hypothesised that nHEV interacts with ITGB1, as a result is subsequently internalised, and trafficked with ITGB1 along the endocytic pathway. First, we used endosomal

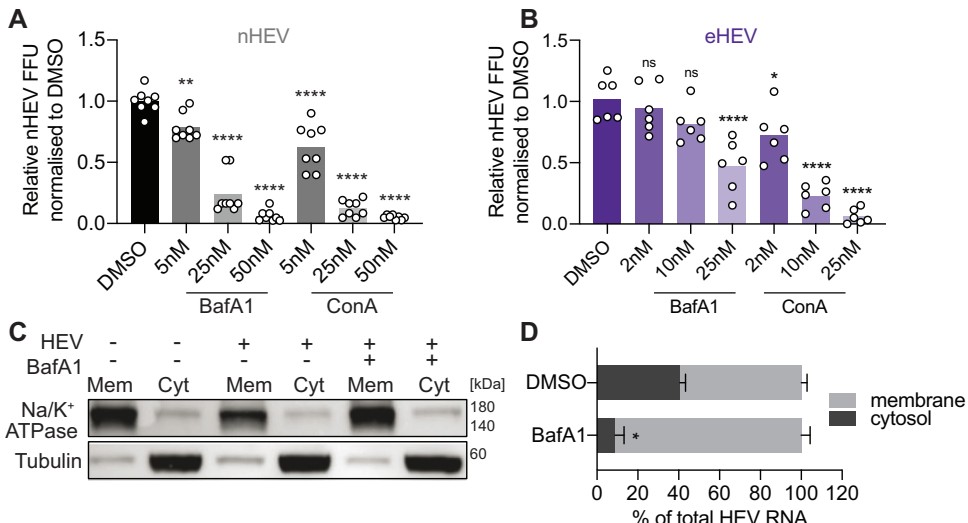

**Fig. 5 | Both nHEV and eHEV entry depend on endosomal acidification.** (**A**) and (**B**) S10-3 cells were treated with indicated concentrations of bafilomycin A (BafA1), concanamycin A (ConA) or DMSO for 30 min prior to infecting with (**A**) nHEV (MOI = 0.1 GE/cell) or (**B**) eHEV (MOI = 5 GE/cell). Drugs and virus were removed after 24 h and HEV infection was quantified by counting ORF2-positive FFUs 5 days post-infection. n = 8 (A) or 6 (B). (**C**) S10-3 cells were treated with 50 nM of bafilomycin A (BafA1) or vehicle control DMSO for 30 min before inoculation with nHEV (MOI = 50 GE/cell). The inoculum was removed 8 h later and replaced with fresh media containing the drug. 24 h later, cells were harvested followed by cell

fractionation to extract membranes and the cytosol. Equal volumes from each fraction were separated by SDS-PAGE and probed by Western blotting against the membrane marker Na/K+ ATPase, and the cytosolic marker tubulin. (**D**) HEV genome copies in each fraction were quantified by RT-qPCR. Shown are percentages of HEV genomes in each fraction in cells treated with BafA1 or DMSO. Data presented as mean ± SD. n = 6 replicates. All data are from two (**C**, **D**) or three (**A**, **B**) independent experiments. Statistical analysis was performed by comparing HEV RNA in cytosolic fractions by unpaired two-tailed Student's t test (**D**) or one-way ANOVA (**A**, **B**). *: $p < 0.05$; **: $p < 0.01$; ****: $p < 0.0001$; ns, non-significant.

acidification inhibitors bafilomycin A (BafA1) and concanamycin A (ConA) and applied them to S10-3 cells 30 min prior to nHEV and eHEV infection (Fig. 5A, B, non-normalised data in Suppl. Figure 4). Both treatments resulted in a dose-dependent decrease in eHEV and nHEV FFU counts 5 days post-infection, in the absence of any drug-induced cell toxicity (Suppl. Fig. 14A). As eHEV entry is more sensitive to these inhibitors[8], we applied lower concentrations for eHEV compared to nHEV to better illustrate the dose-dependent effect on eHEV infection. We verified that the drug treatments did not affect HEV genome replication (Suppl. Fig. 14B) and further confirmed their effect on nHEV infection in HepG2/C3A cells as well as in primary human hepatocytes (Suppl. Fig. 14C, D).

To further corroborate our findings, we separated cell membranes from the cytosol in BafA1-treated and untreated HEV-infected cells (Fig. 5C) and found that the BafA1 treatment led to an enrichment of detected HEV genomes in the membrane fraction, 24 h post-infection (Fig. 5D). Since the separation of membrane and cytosolic fractions is imperfect (Fig. 5C), a small percentage of HEV RNA was still detected in the cytosol, despite the BafA1 treatment.

We also confirmed our findings using our entry assay based on HEV genome detection and capsid staining. We found a significant increase in capsid-associated genome-positive particles for both nHEV and eHEV upon endosomal inhibitor treatment compared to DMSO-treated cells (Fig. 6A, B), suggesting that uncoating did not occur in the presence of the inhibitors.

Next, we wanted to investigate the specific endosomal compartments through which eHEV and nHEV particles are trafficked. To this end, we used siRNAs to knock down Rab5a, Rab7, and Rab11 (Suppl. Fig. 15), which are typical markers of early, late, and late recycling endosomes, respectively. We found that knockdown of Rab11, but not Rab5a, resulted in a significant increase in capsid-associated nHEV genomes detected by RNA-FISH (Fig. 6C). In contrast, knockdown of Rab5a, but not Rab11, resulted in a significant increase in capsid-associated genomes of eHEV particles (Fig. 6D). Interestingly, Rab7 knockdown affected both eHEV and nHEV uncoating (Fig. 6C, D).

Furthermore, we analysed if HEV particles could be detected in the respective endocytic compartments. To this end, we generated S10-3 cells ectopically expressing EGFP-tagged Rab5a, Rab7, and Rab11 and analysed the cells by high-resolution microscopy after allowing binding and internalisation of eHEV and nHEV particles (Fig. 6E-G). First, we found more eHEV than nHEV particles within Rab5a positive vesicles (Fig. 6E). In contrast, we found nHEV, but less eHEV particles, colocalising with ITGB1 in Rab11-positive endosomes (Fig. 6F), supporting our hypothesis that the interaction with ITGB1 sorts nHEV, but not eHEV particles, into Rab11-positive vesicles, with Rab11 being a hallmark of recycling endosomes. Further confirming the relevance of the recycling endosome in nHEV entry, we found that siRNA-mediated downregulation of other components of the recycling endosome, such as, EHD1 and MICAL-L1, resulted in less efficient nHEV uncoating (Suppl. Fig. 16). Finally, we detected colocalisation of both nHEV and eHEV particles with the late endosomal marker Rab7 (Fig. 6G). Altogether these results suggest that while eHEV traffics through the classical endocytic pathway via Rab5, nHEV appears to rely on the Rab11-positive recycling endosome with both particle forms appearing to converge on the Rab7-positive late endosome.

### HEV uncoating in the lysosome requires cathepsin activity

Finally, we investigated whether eHEV and nHEV traffic until they reach the final destination of endocytic cargoes, the lysosome. Since HEV particles are enterically transmitted and exposed to a highly acidic pH in the gut, we hypothesised that additional triggers are required for HEV uncoating. Lysosomal cathepsins have been shown to be involved in the entry of many viruses[36-38], including HEV[43]. Therefore, we speculated that capsid processing by cathepsins might be important for HEV capsid uncoating and subsequent genome release. As initial evidence, we generated stable S10-3 cells expressing the EGFP-tagged lysosomal marker LAMP1 and we found nHEV and eHEV particles in LAMP1-positive vesicles, 7 h and 10 h post-internalisation, respectively (Fig. 7A). We then treated cells with the cathepsin inhibitor E64 either before or after nHEV and eHEV infection and quantified FFUs 5 days post-infection. We found that the E64 inhibitor significantly reduced

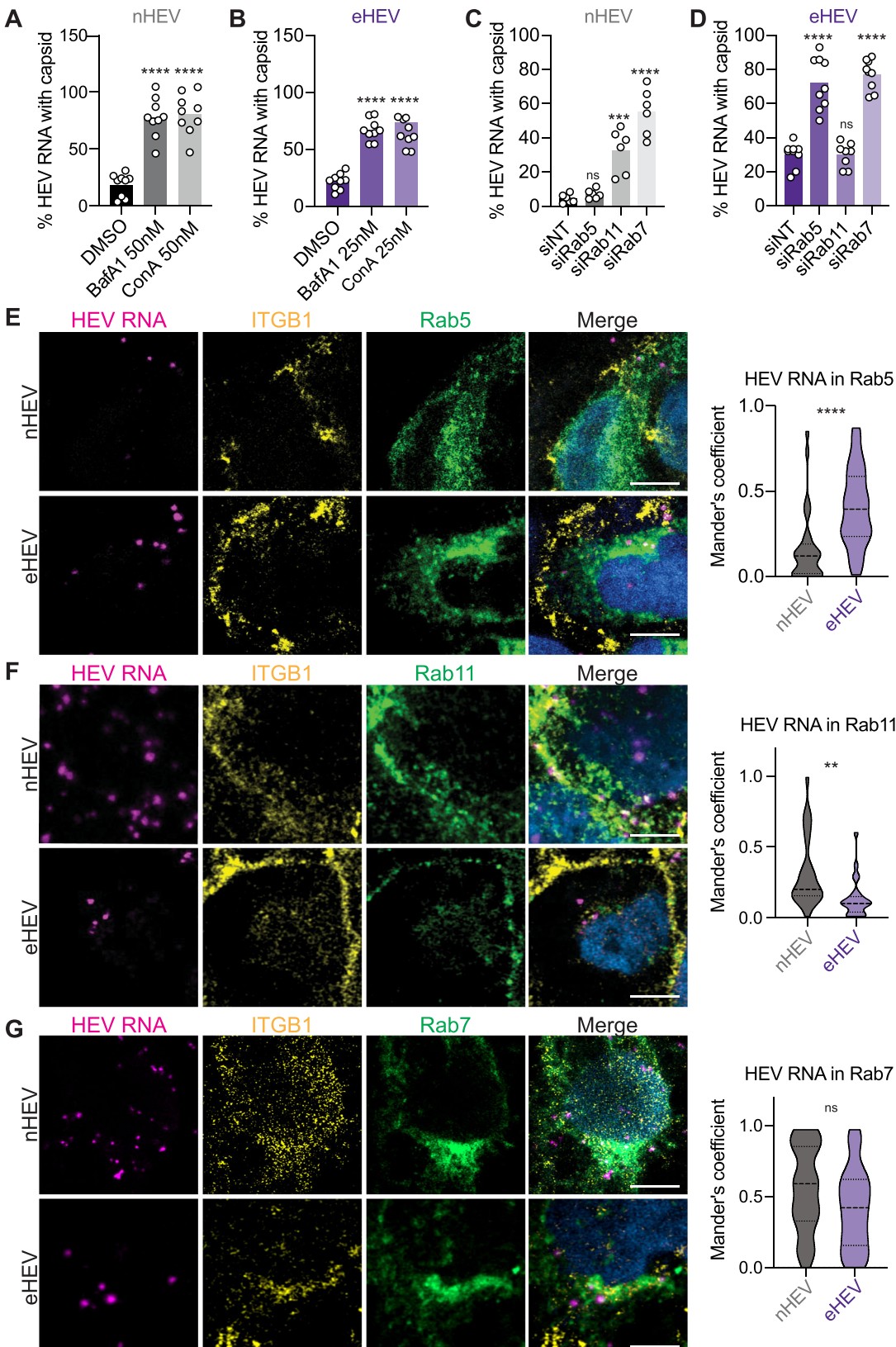

nHEV and eHEV infection when applied before infection. In contrast, the application at 24 h post-infection had no effect on either eHEV or nHEV infection (Fig. 7B, C, non-normalised data in Suppl. Figure 4).

Next, we confirmed the effect of E64 on nHEV and eHEV entry using our RNA-FISH based entry assay. We found a significant increase in capsid-associated genomes of both nHEV and eHEV particles upon E64 treatment compared to DMSO-treated cells (Fig. 7D, E). Furthermore, we observed entrapment of HEV capsid and colocalisation with the genome in LAMP1-positive lysosomes 24 h post-infection when treated with E64 (Fig. 7F, G), suggesting unsuccessful uncoating upon cathepsin inhibition. Of note, we still detected faint capsid signals by microscopy 24 h post-infection (Fig. 4B), which likely corresponded to

**Fig. 6 | nHEV and eHEV particles use different endocytic pathways. (A)** and **(B)** S10-3 cells were treated with BafA1, ConA or DMSO for 30 min before inoculation with **(A)** nHEV (MOI = 30 GE/cell) and **(B)** eHEV (MOI = 20 GE/cell). The inoculum was removed 8 h later and replaced with fresh media containing the drugs. 24 h later, cells were fixed and HEV capsid and genomes were detected by immunofluorescence staining and RNA-FISH using the ORF1 probe, respectively, and quantified using CellProfiler. Shown are the percentages of HEV genomes colocalising with HEV capsids out of the total number of detected genomes per cell. n = 9 microscope fields. **(C)** and **(D)** S10-3 cells were transfected with siRNAs directed against Rab5, Rab7, Rab11 and a NT-control. 48 h post-transfection, cells were inoculated with **(C)** nHEV (MOI = 30 GE/cell, n = 6 microscope fields) or **(D)** eHEV (MOI = 20 GE/cell, n = 9 microscope fields) and incubated for 8 h at 37 °C. 24 h post-inoculation, analysis was performed as in (A). **(E-G)** S10-3 cells ectopically expressing **(E)** EGFP-Rab5 or **(F)** EGFP-Rab11 or **(G)** EGFP-Rab7 were inoculated with nHEV (MOI = 30 GE/cell) or eHEV (MOI = 20 GE/cell) for 2 h or 6 h on ice, respectively, and incubated for **(E)** 1 h, **(F)** 15 min, **(G)** 2 h (nHEV) or 8 h (eHEV) at 37 °C, respectively. Cells were stained against ITGB1 (yellow). HEV genomes (magenta) were visualised as described in (A). Images are representatives of n = 6 microscope fields. Scale bar = 5 μm. All data are from three independent experiments. Statistical analysis was performed by one-way ANOVA **(A-D)** or unpaired two-tailed Student's t test **(E–G)**. **: $p < 0.01$; ***: $p < 0.001$ ****: $p < 0.0001$; ns, non-significant.

the detection of partially degraded capsids within the lysosome. Thus, both nHEV and eHEV particles appear to uncoat in the lysosome and to require lysosomal cathepsins for genome release.

## Discussion

The early stages of the HEV life cycle are poorly understood. One obvious reason for this is the lack of suitable methods to study them separately from the later steps of the life cycle. Here, we used an RNA-FISH-based assay to describe authentic nHEV and eHEV cell entry steps, from the interaction with potential surface receptors to trafficking through the endocytic pathway. We provide compelling evidence that ITGB1 acts as a co-factor for nHEV but not eHEV entry. We further found that the two particle forms are differentially trafficked along the endocytic pathway and that lysosomal cathepsin activity is critical for particle uncoating of both forms (Fig. 7H).

A previous study using PLC/PRF/5 hepatoma cells proposed ITGA3 as an essential host factor for nHEV cell entry[14]. To our surprise, we were barely able to detect ITGA3 in other, highly HEV-permissive hepatoma cell lines (Fig. 1A, B). These analyses also revealed that the expression of individual α-integrins was generally variable, while ITGB1, which can pair with many α-integrins, was expressed across all cell lines tested. Indeed, our data showed that ITGA2, rather than ITGA3, may be involved in nHEV infection of Huh-7-based cell lines, which are widely used for HEV infection studies.

Therefore, we believe that a specific α-integrin is not the determinant for nHEV entry. It is likely that nHEV uses different α-integrins in a cell-type dependent manner, whereas ITGB1 is the critical integrin that mediates nHEV entry universally. In support of this, other viruses have also been reported to use different integrin β1 heterodimers for cell entry. For example, integrins α2β1 and α6β1 can both support entry of human cytomegalovirus[44]. Both integrins αvβ1[45] and α2β1[46] were reported to mediate entry of SARS-CoV2. To support the role of ITGB1 as the main factor, we showed that ITGB1 modulation affects nHEV entry across different cell types. While the HEV capsid lacks the classic ITGB1 ligand motif, we have identified a reverse DGR motif in the protruding P-domain of ORF2 that could mediate binding to the different integrin β1 heterodimers.

The expression of integrin heterodimers on the cell surface is tightly and dynamically regulated. When one subunit of integrins is downregulated or impaired, other subunits are known to compensate for their functions[47]. This compensatory mechanism may explain the differential dependency of nHEV on the α-integrin subunits and the relatively mild phenotype we observed in ITGA2 and ITGB1 knockout cells (Fig. 1). Critically, ITGB3 and ITGB5 heterodimers, similar to ITGB1 heterodimers, also bind to their ligands via their RGD motif[48] and herpes simplex virus 1 (HSV-1), for example, has been reported to use different integrin pairs as interchangeable receptors[49]. This ability allows the virus to adapt to alternative pathways in different cell types, thereby broadening its range of cellular targets for infection. Indeed, the interaction with a promiscuous factor such as ITGB1 could explain the broad tissue and species tropism of the HEV strain used in this study (reviewed in[50]). As the present work was limited to the use of a zoonotic strain, future studies should include strains with a narrower

tropism, such as the HEV-1 and −2 genotypes that are restricted to human infection.

While ITGB1 is ubiquitously expressed, the α-integrins appear to be more tissue specific, e.g. α3 and α6 in epithelial cells[51] and α10 in chondrocytes[52] (reviewed in[53]). It would therefore be interesting to investigate whether ITGB1 heterodimerises with specific α-integrin partners that could be critical in mediating HEV entry into different tissues, such as the brain[54] or intestine[55], which have been described as potential reservoirs for HEV infection in chronic patients (reviewed in[56]).

In line with this, we have found that downregulation of ITGB1 in the lung cell line A549 and the intestinal cell line Caco-2 impaired nHEV but not eHEV infection (Suppl. Figure 6B, C), further supporting that the role of ITGB1 in HEV infection is not limited to hepatocytes. Since the intestinal environment favours the removal of the quasi-envelope, it is possible that HEV particles re-enter the body through naked particles released with the bile into the intestine. Therefore, the interaction between nHEV and ITGB1 may play a prominent role in HEV transmission along the gut-liver axis.

Schrader and colleagues previously identified the epidermal growth factor receptor (EGFR) as an nHEV cell entry factor in hepatocytes[12]. Interestingly, both β1 and β3 integrins have been shown to associate with EGFR, activate EGFR in a ligand-independent manner (e.g. activate EGFR through integrin binding)[57] and synergistically enhance EGFR signaling[58]. Several viruses have been reported to exploit EGFR and integrins simultaneously in their life cycle, with one study showing that integrin-EGFR coordination is essential for human cytomegalovirus entry[59]. Therefore, future studies should investigate whether HEV entry requires co-signalling or coupling of EGFR with ITGB1.

In contrast to many other cell surface receptors that undergo synchronised ligand-induced internalisation and degradation, integrins are constantly recycled in cells[36]. Following ligand binding, integrins are internalised and quickly and predominantly recycled back to the membrane, whereas integrin degradation is rather slow (reviewed in[60]).

Foot-and-mouth disease and vaccinia viruses have been shown to traffic through the recycling endosome in an integrin-dependent manner[61,62]. Our data suggest that the HEV capsid interacts directly with a cell-type dependent integrin β1 heterodimer and activates its internalisation, possibly mediated by the recruitment of pFAK to Rab11+ recycling endosomes. We observed nHEV but not eHEV colocalisation with ITGB1 on the cell surface (Fig. 3E, Suppl. Fig. 9A, B) as well as in Rab11+ endosomes (Fig. 6F). By contrast, in the absence of interaction with ITGB1, eHEV particles seem to traffic through the classic endocytic route via Rab5+ early endosomes. In fact, Rab5 is encoded by three isoforms (Rab5a, b, c) that cooperate in the regulation of endocytosis in eukaryotic cells[54]. Rab5a selectively regulates growth factor receptor trafficking, whereas Rab5c appears to regulate cell motility and cytoskeletal dynamics[55]. In addition, receptor complexes involving ITGB1 have been shown to be recycled to the surface mainly in dependence on Rab5c[63]. As Rab5a is the most abundantly expressed isoform, we used an siRNA targeting Rab5a in our study and found that the downregulation affected eHEV but not nHEV entry. We

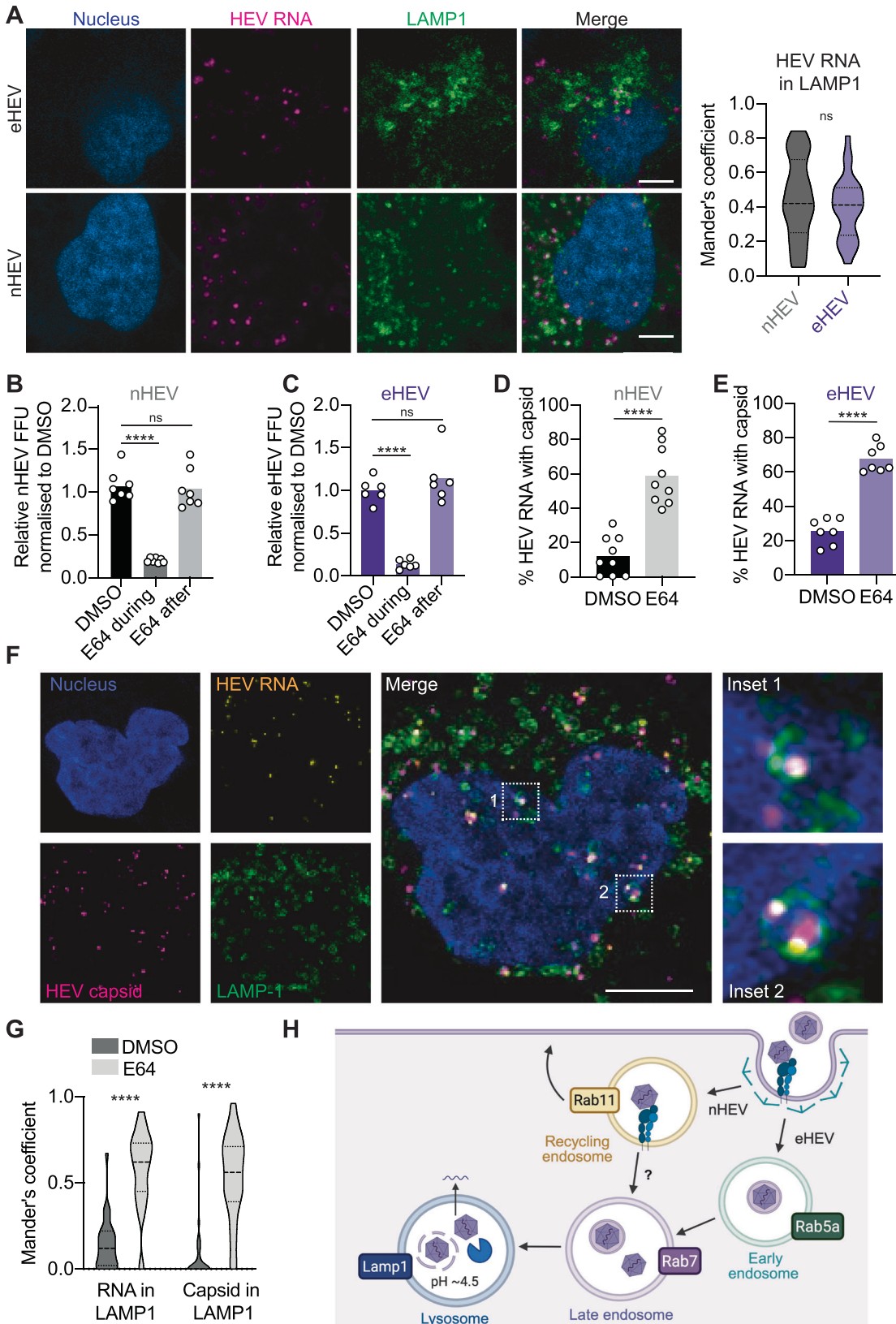

also found that eHEV, but not nHEV, colocalised with EGFP-Rab5a (Fig. 6E). Therefore, we propose that the different dependencies on ITGB1 may be the determinants that dictate the different trafficking pathways of the two HEV particle types. It would be interesting to investigate whether a fraction of nHEV particles is recycled back to the membrane and requires re-internalisation with ITGB1. Furthermore,

the potential involvement of Rab5c in trafficking of nHEV particles should be investigated. The RNA-FISH based entry assay developed in this study could be used in the future to unravel these details of early HEV cell entry.

Following endocytic trafficking, we found nHEV particles in LAMP1-positive compartments (Fig. 7A), which are likely to be

**Fig. 7 | nHEV and eHEV particles require lysosomal cathepsin activity for cell entry.** (**A**) S10-3 LAMP-1-GFP cells were inoculated with eHEV (MOI = 20 GE/cell) or nHEV (MOI = 30 GE/cell) for 6 h or 2 h on ice and incubated at 37 °C for 10 h or 7 h, respectively. Genomes (magenta) were detected by RNA-FISH using the ORF1 probe. Scale bar = 5 μm. (**B**) and (**C**) S10-3 cells were treated with cathepsin inhibitor E64 or DMSO and infected with (**B**) nHEV (MOI = 30 GE/cell, n = 7) or (**C**) eHEV (MOI = 20 GE/cell, n = 6). E64 was added with virus for 24 h ("during"), or 24 h post-infection and throughout the course of infection ("after"). Infectivity was assessed 5 days post-infection. (**D**) and (**E**) S10-3 cells were treated with E64 or DMSO and infected with (**D**) nHEV (MOI = 30 GE/cell, n = 9) or (**E**) eHEV (MOI = 20 GE/cell, n = 7). 8 h later, the inoculum was replaced with fresh media containing drugs. 24 h post-inoculation, HEV capsid were detected by staining and genomes as in (**A**) and quantified using CellProfiler. (**F**) Maximum projections of E64-treated S10-3 LAMP-1-GFP cells inoculated with nHEV (MOI = 30 GE/cell). 24 h later, capsids (magenta)

and genomes (yellow) were detected as in (**C**). Representative of n = 6 microscope fields. (**G**) S10-3 LAMP-1-GFP cells were treated with E64 or DMSO 30 min prior to inoculation with nHEV (MOI = 30 GE/cell). After 8 h at 37 °C, inoculum was replaced with fresh media containing drugs. Genomes were detected and analysed as in (**A**). (**H**) Proposed working model on HEV cell entry. The interaction of nHEV with ITGB1 triggers internalisation through Rab11+ recycling endosomes, while eHEV is routed into Rab5a+ early endosomes. Both particles traffic through Rab7+ late endosomes and reach Lamp1+ lysosomes. The capsid and envelope are degraded by lysosomal cathepsins, allowing the release of viral genomes into the cytosol through an unknown penetration mechanism. This figure was created in BioRender. Dao Thi, V. (2025). https://BioRender.com/f6k0uqw. All replicates are from three independent experiments. Statistical analysis was performed by unpaired two-tailed Student's t test (**A**, **D**, **E**) or one-way ANOVA (**B**, **C**, **G**). ****: $p < 0.0001$; ns, non-significant.

endolysosomes. A number of receptors and viruses have been reported to enter cells via the recycling endosome and ultimately traffic through the late endosomes and/or lysosomes[64–66]. A possible link between the recycling endosome and the endolysosome is autophagosome formation. Rab11 has been proposed to regulate the fusion of MVBs with autophagosomes[67], and LAMP1 is known to be distributed among autophagic organelles[68]. In addition, Rab7 has been shown to be important for the autophagic pathway and the maturation of autophagosomes[69,70]. It is possible that the autophagy machinery could promote the recruitment of molecular motors to the ruptured endosome[71] and assist in the exit of the HEV genome from the endosomes. Therefore, future studies should aim to further investigate the role of the autophagosome in HEV cell entry.

Our data suggest that both nHEV and eHEV traffic through the lysosome and require the activity of cathepsins for successful infection. Viral capsids can be processed by cathepsins, which leads to the exposure of peptides that harbour membrane penetration activity in order to achieve endosomal escape[72–74]. In agreement, we recently found HEV capsid processing by cathepsin L[43] and further work should aim at determining the underlying mechanism leading to nHEV and eHEV escape from the endosome.

Recently, Rab5 has been identified as host factor of HEV replication[75] and Rab11 has been shown to play a role in HEV replication and assembly[76]. Indeed, for many viruses, entry and assembly/secretion pathways can be intricately linked (e.g. β-coronaviruses[77]). These observations suggest that end-point HEV infection assays are not suitable to assess the role of proteins, e.g. by knockdown or knockout during cell entry and they highlight the need to use an assay that clearly separates the cell entry steps from the later stages of the viral life cycle.

Here, we studied HEV particles by detecting RNA and/or capsid within a timeframe of 8-24 h post-infection. The application of a specific inhibitor showed that within this timeframe, we indeed investigated only the early steps of the HEV life cycle, independent of genome replication (Suppl. Figure 7E). Therefore, our approach ensures that the involvement of cellular factors such as Rab proteins is specifically linked to HEV entry and not to subsequent stages of the viral life cycle such as replication and/or progeny assembly.

In contrast to nHEV particles, eHEV particles, which are unlikely to contain virus-encoded proteins exposed on the quasi-envelope, do not appear to interact with ITGB1 (Fig. 3). This is in contrast to HAV, for which ITGB1 depletion reduced the cell entry of both naked and quasi-enveloped particles[17].

Both eHAV and eHEV particles acquire their quasi-envelope by budding into MVBs, and a previous study has shown that ITGB1 heterodimers may be involved in extracellular vesicle uptake[78]. Therefore, ITGB1 depletion could play a direct role in the uptake of quasi-enveloped viral particles, even in the absence of a direct capsid interaction.

A major reason for the different dependencies observed could be the different cell types used: While Rivera-Serrano and colleagues used the cervical cancer cell line HeLa to show the importance of ITGB1 for both naked and quasi-enveloped HAV particles[17], we mainly used hepatoma cell lines in our study. The uptake of extracellular vesicles and/or quasi-enveloped particles is likely to be mediated by different integrins depending on the cellular background, as e.g. ITGB3 but not ITGB1 is critical for EV uptake in the breast cancer cell line MDA.MB.231[79,80].

Consistent with this, the dependence of eHAV particles on the TIM-1 receptor appears to be cell type dependent, as a significant decrease in eHAV was previously observed in TIM1-KO Vero cells, but not in TIM1-KO Huh-7.5 cells[81].

Many open questions remain about the cell entry pathways used by both nHEV and eHEV particles. Many viruses complete the entry process within an hour[82–84], whereas nHEV and eHEV particles appear to require more than 12 h to uncoat. Future studies should aim to further elucidate the molecular mechanism in a time-resolved manner of each step. In this manuscript, we have laid the groundwork and provided a novel entry assay to enable such studies. A better description of the HEV cell entry steps could lead not only to the development of therapeutic interventions, but also to a better understanding of HEV tropism and extrahepatic manifestations.

## Methods
### Cell culture
The human hepatoma cell lines S10-3 (a kind gift from Suzanne Emerson, NIH) and HepG2/C3A and their derivatives were cultured in Dulbeccos's Modified Eagle Medium (DMEM, Gibco) + GlutaMAX-I supplemented with 10% fetal bovine serum (FBS, Capricorn), 1% penicillin/streptomycin (P/S) (Gibco) here referred to as complete DMEM (cDMEM). Cell lines were validated by phenotypic screening and confirmed to be mycoplasma-free using a PCR detection kit (Abcam). All cells were maintained at 37 °C in a 95% humidity and 5% $CO_2$ atmosphere.

### Production and purification of naked and enveloped HEV particles
nHEV and eHEV particles were harvested from S10-3 cells electroporated with in vitro transcribed HEV GT3 Kernow C1 P6 (GenBank accession number: JQ679013.1) RNA, 7 days post-electroporation. In brief, nHEV particles were collected from the cell lysate through four freeze-thaw cycles, and eHEV particles were harvested from the filtered cell culture supernatant. The HEV particles were then concentrated by layering on top of a 20% w/v sucrose cushion and ultracentrifuged at 100,000 x g using a SW 32 Ti Swinging-Bucket Rotor for 3 h at 4 °C. The resulting pellet was resuspended in PBS and further purified with help of a continuous gradient, which was prepared by layering 2.5 ml of 60%, 40%, 25%, and 15% Opti-prep™ (Sigma) (v/v) each in a Thinwall

Ultra-Clear Tube (Beckman). The gradient was placed horizontally for 1 h to allow mixing of the gradient and kept overnight at 4 °C. The next day, concentrated virus was layered on top of the gradient and centrifuged at 126,000 x g for 16 h at 4 °C using a SW 41 Ti Swinging-Bucket Rotor. Twelve individual 1 ml gradient fractions were manually collected. The refractive index of each fraction was measured using a digital handheld refractometer (DR201-95, Krüss). Fractions with a density ranging from 1.05 to 1.15 g/cm$^3$ were pooled for eHEV and from 1.2 to 1.25 g/cm$^3$ for nHEV infection experiments. All infection experiments were performed using density-purified eHEV and nHEV particles. Prior to infection, the gradient-purified virus was subjected to buffer exchange using the Pur-A-Lyzer™ Maxi Dialysis Kit (20k MWCO) (Sigma) at 4 °C overnight to replace the iodixanol with PBS.

## HEV infection and neutralization assays

$3 \times 10^4$ S10-3 cells were seeded in a well of a 48-well plate and infected with nHEV or eHEV in MEM (Gibco) supplemented with 10% FBS and 1% P/S, the next day. The inoculum was removed after 8 h and replaced with cDMEM. 5 days post-infection, the cells were fixed in 4% PFA (Electron Microscopy Sciences) for 15 min and permeabilised with methanol at −20 °C for 20 min. The cells were then blocked with 10% goat serum for 1 h at room temperature, immunostained with an ORF2 antibody (1E6 1:1000, Millipore) at 4 °C overnight followed by anti-mouse Alexa-594 (1:1000, Thermo Fisher) staining for 1 h at room temperature. All infection assays were carried out at 37 °C. For RGD treatment, cells were pre-incubated with the RGD-containing peptide GRGDNP (Santa Cruz) or a control peptide with scrambled RGD sequence GRADSP (Enzo) at indicated concentrations for 15 min followed by HEV infection as described above. For all other drug treatments, cells were pre-treated for 30 min prior to HEV infection. Bafilomycin A (Sigma), Concanamycin A (Biomol) and E64 (Selleckchem, 25 µM) were used at the indicated concentrations. For all infectivity assays, images of entire wells were taken with a Zeiss Cell-Discoverer 7 microscope and the number of FFU was counted manually. For neutralisation, $4 \times 10^4$ S10-3 cells were seeded on a 10 mm coverslip in a well of a 48-well plate. Gradient-purified nHEV was neutralised with convalescent patient-serum or HEV-negative patient serum at 1:1000 for 1 h at 37 °C. The cells were then inoculated with serum-treated or untreated nHEV for 6 h at 37 °C. The use of the patient-serum was approved by the Ethics Committee of the Albert-Ludwigs-University Freiburg (474/14, 201/17, 486/19), and written informed consent was obtained from all blood donors before enrolment in the study.

## Generation of HEV mutant

The plasmid pBlueScript SK(+) carrying the DNA of the full-length genome of HEV GT3 Kernow C1 P6 (GenBank accession number: JQ679013.1) was used as a template. Mutation of the D522 residue to E in ORF2 was generated by site directed mutagenesis. Nucleotide mutations were introduced by overlap extension PCR using the Phusion polymerase (New England Biolabs, NEB) with the primers listed in Table 1 (ORF2 full length F, ORF2 overlap F, ORF2 overlap R and ORF2 full-length R). Restriction enzyme digestions and ligation were performed subsequently, and the mutation was verified by DNA sequencing.

## Generation of integrin knockout cell lines

S10-3 and Huh-7 cells were transduced with lentiviruses carrying the lentiCRISPRv2 (Addgene #49535) plasmid encoding Cas9 and sgRNAs. Two days post-transduction, cells were selected with puromycin (2 µg/ml) and expanded stepwise. For single-cell knockout (KO) screening, bulk KO cells were seeded onto pre-plated Lenti-X helper cells in 96-well plates (0.5 cell/well) in cDMEM with 20% FBS. Selection media with puromycin was introduced 7–10 days post-seeding and refreshed every two days until selection was complete. Surviving single-cell

clones were expanded and characterized for integrin expression via immunofluorescence and Western blot analysis. Guide RNA sequences against ITGB1 and ITGA2 (sequences and target regions listed in Table 1) were designed using the online tool Synthego (https://design.synthego.com/#/).

## Establishment of stable cell lines

The ITGB1 gene (GenBank accession number: NM_002211.4) was cloned into the lentiviral expression plasmid pWPI (Addgene #12254) and the EGFP-Rab5a, EGFP-Rab7, EGFP-Rab11 and EGFP-LAMP1 genes[85] into the doxycycline-inducible lentiviral expression plasmid pTRIPZ (Thermo Fisher). Lentiviruses were produced by transfecting HEK293T cells with plasmids encoding VSV-G, HIV gag/pol proteins, and the transgene using the JetPRIME reagent (Polyplus) according to the manufacturer's protocol. Lentiviruses were harvested 48 h post-transfection. S10-3 ITGB1 KO cells were transduced with pWPI-ITGB1 and selected in cDMEM supplemented with 400 µg/ml G418 (Invivogen). S10-3 WT cells were transduced with pTRIPZ-EGFP-Rab5a/7/11 or EGFP-LAMP1 and selected in cDMEM supplemented with 2 µg/ml puromycin (Invivogen). Transduced cells were treated with 2 µg/ml doxycycline (Sigma) for 48 h to induce EGFP-Rab5a/7/11 or EGFP-LAMP1 expression.

## Quantitative (real-time) reverse transcription-PCR

RNA was extracted from cell lysate or purified virus with Trizol reagent (Sigma) following the manufacturer's recommended protocol. cDNA was synthesized using the iScript cDNA Synthesis Kit (BioRad). HEV RNA GEs were quantified using the SYBR Green Master mix (BioRad) and primers targeting the HEV genome (Table 1). RPS11 gene was used as a housekeeping control. The qPCR reaction was carried out in a BioRad CFX Maestro instrument.

## siRNA reverse transfection

SMARTpool siRNAs (Dharmacon) were individually added to each well of a 48-well plate containing 25 µl OptiMEM and 1 µl Lipofectamine™ RNAiMAX Transfection Reagent (Thermo Fisher) at a final concentration of 100 nM. After 5 min incubation at room temperature, $5 \times 10^4$ S10-3 cells were added to each well in 250 µl cDMEM. Medium was changed after 24 h. 48 h post-transfection, the cells were inoculated with HEV or harvested for western blot analysis.

## In situ labelling of viral RNA and immunofluorescence staining

Infected S10-3 cells seeded onto coverslips were fixed in 4% PFA and permeabilised in 0.1% Triton-X100 (Sigma). For co-detection of RNA and capsid, RNAscope® Fluorescent Multiplex Kit version 1 (ACDBio) was used according to the manufacturer's protocol. The positive strand of HEV RNA was targeted by the ORF1 probe (ACDBio, Cat No. 579831) or ORF2 probe (ACDBio, Cat No. 586651). Subsequently, cells were blocked in 5% goat serum followed by immunostaining with 1E6 (Millipore, 1:400) or ITGB1 antibodies (Santa Cruz, 1:100) at room temperature for 1 h. The respective Alexa Fluor-conjugated secondary antibodies (Thermo Fisher) were used at 1:1000 diluted in 5% goat serum and incubated at room temperature for 1 h. Finally, the coverslips were mounted using the ProLong™ Glass Antifade Mountant (Thermo Fisher) and cured for at least 24 h in the dark. RNAscope® Fluorescent Multiplex Kit version 2 (ACDBio) was used for detection of only RNA. Cells were fixed and permeabilised as described above, followed by $H_2O_2$ treatment for 10 min at room temperature before proceeding with the RNAscope protocol.

## Western blot

Cells were lysed in RIPA lysis buffer (Thermo Fisher) on ice for 30 min and the protein concentration was determined using the Pierce™ BCA protein assay kits (Thermo Fisher) to ensure equal loading. A total of 20 µg of proteins were mixed with 6x SDS loading dye

**Table 1 | List of oligonucleotides used in this study**

| Oligonucleotide | Target | Sequence (5'–3') | Ref |
|---|---|---|---|
| ORF2 full-length F | | TGTACCTGATGTTGATTCACGTGGTGCTATTC | |
| ORF2 overlap F | Position 522 in ORF2 | GTTACTTTGGAGGGTCGCCCCCTTA | |
| ORF2 overlap R | Position 522 in ORF2 | TAAGGGGGCGACCCTCCAAAGTAAC | |
| ORF2 full-length R | | GGCACGGAAGGAATTAATTAAGACTCCCGG | |
| HEV F | HEV | GGTGGTTTCTGGGGTGAC | 84 |
| HEV R | | AGGGGTTGGTTGGATGAA | |
| RPS11 F | RPS11 | GCCGAGACTATCTGCACTAC | 7 |
| RPS11 R | | ATGTCCAGCCTCAGAACTTC | |
| sgRNA ITGB1 | ITGB1 exon 7 (out of 16) | AGAATTTCAGCCTGTTTACA | |
| sgRNA ITGA2 | ITGA2 exon 2 (out of 30) | TGTTGTTTGGCCTACAATGT | |

containing 10% 2-mercaptoethanol (VWR Life Sciences) and boiled at 100 °C for 10 min before loading. Proteins were then transferred to polyvinylidene difluoride (PVDF, G-Biosciences) membranes by wet blotting using standard methods. The membranes were blocked with 5% milk/0.1% Tween-20 in PBS (PBS-T). Antigens were stained with the indicated antibodies in 5% milk: mouse α-ITGB1 1:500 (Santa Cruz); rabbit α-ITGA2 1:500 (Abcam); rabbit α-ITGA3 1:500 (Millipore); rabbit α-ITGA5 1:500 (Abcam), mouse α-ITGA6 1:500 (St.John's Laboratory); mouse α-actin (Sigma) 1:4000; mouse α-FAK (Santa Cruz) 1:500; rabbit α-Rab 7 (Abcam) 1:1000; rabbit α-Rab 11 (Abcam) 1:1000; goat α-Rab 5 (antibodies-online) 1:1000; α-tubulin (Sigma) 1:2000 and α-Na/K+ ATPase (Millipore) 1:1000, followed by staining with corresponding secondary antibodies conjugated with HRP (Jackson ImmunoResearch). Membranes were imaged with Pierce™ Enhanced Chemiluminescence (ECL) Western Blotting Substrate and images were acquired using ChemoStar Touch ECL & Fluorescence Imager (Intas).

### Confocal microscopy and image analysis

Multichannel z-series with a z-spacing of 10 μm, or single slice confocal images were acquired using a Leica LIGHTNING SP8 or a Zeiss Airyscan LSM900 confocal microscope, as indicated in the figure legend. A 63× oil immersion objective was used for all images. For quantification of HEV genomes per cell during entry and percentages of HEV genomes associated with capsid, maximum projections of full z-series were used. The genomes per cell were estimated by dividing the total number of detected genomes by the number of nuclei in a frame. Partially detected nuclei were excluded from the analysis. Images were processed using the Zen software and inspected manually before quantifications using CellProfiler. Colocalisation of fluorescence images of single slices by confocal microscopy was quantified by evaluating Manders' coefficient using the "Colocalisation" module in Zen 2.1. 20 − 30 cells were analysed for each experimental condition and presented as violin plots.

### Sample preparation and Liquid chromatography–mass spectrometry (LC−MS/MS) analysis

Nano-flow LC-MS/MS analysis and sample preparation was described previously[86]. Briefly, for the full proteome characterisation of the liver cancer cell lines, cells were lysed in an adapted RIPA buffer. The BCA Assay (Pierce) was used to estimate protein concentration. Protein digestion and clean-up were performed using an adapted version of the automated paramagnetic bead-based single-pot, solid-phase-enhanced sample-preparation (Auto-SP3) protocol on a Bravo liquid handling platform (Agilent). For protein digestion, and samples were incubated overnight at 37 °C. After digestion, the recovered peptides were dried by vacuum centrifugation (150 x g at 45 °C), and stored at −80 °C until use. For nano-flow LC−MS/MS analysis, an Ultimate 3000

HPLC (Thermo Fisher) was coupled to an Orbitrap Exploris 480 mass spectrometer (Thermo Fisher). After loading, peptides were separated using a 141 min gradient from 8 to 38% of buffer B (0.1% FA, 80% ACN in MS-H$_2$O) at a 300 nL/min flow rate. The Orbitrap Exploris 480 mass spectrometer was operated in data-independent mode (DIA), with an m/z range of 350-1400.

### Ex vivo RNA-FISH analysis of immobilised particles

15-well μ-slide angiogenesis dishes (ibidi) were coated with 30 μl/well polyethyleneimine (PEI; 1 mg/ml) for 30 min at room temperature and washed with PBS. Gradient purified nHEV particles were incubated in PBS on PEI-coated chamber slides for 1 h at 37 °C. Subsequently, samples were washed with PBS, fixed in 4% PFA and viral capsids were permeabilised with 0.1% Triton X-100[87]. In situ labelling of viral RNA followed by immunofluorescence of the ORF2 antibody 1E6 on immobilised particles was then carried out as described previously.

### Proximity ligation assay

S10-3 WT or KO cells were inoculated with nHEV WT or D522E mutant for 2 h on ice to allow particle binding followed by 5 min incubation at 37 °C. Cells were fixed with 4% PFA and permeabilised using Triton X-100. Primary antibodies targeting ORF2 (rabbit, a kind gift from Rainer Ulrich, FLI, 1:500) and ITGB1 (mouse, Santa Cruz, 1:100) were incubated for 1 h at room temperature followed by proximity ligation assay (PLA) according to the manufacturer's protocol (Duolink in Situ PLA Kit).

### Interaction prediction between ORF2 and ITGB1 using AlphaFold

Interaction between ORF2 (Kernow-C1 P6) and human ITGB1 (UniprotID: P05556) was predicted using an in-house implementation of ColabFold[88] employing AlphaFold-Multimer 2.3 weights[89] and a multiple sequence alignment generated with the MMSEQS2 webserver[90]. Given that AlphaFold demonstrates improved performance for smaller regions in predicting interactions[91], we focused on monomeric and dimeric forms of the ORF2 P-domain (ORF2-P). We tested interactions with full-length ITGB1 and a truncated ITGB1 spanning residues 139-380 (numbered from the peptide signal). For each prediction, we used three different seeds to generate fifteen models, facilitating the reproducibility of results and increasing statistical robustness.

Using models with low Predicted Aligned Error (PAE), amino acid residues at the ORF2P-ITGB1 interface were identified for alanine mutagenesis. The impact of these mutations was predicted by calculating the average PAE (PAEinter) between ORF2-P and ITGB1139-380, setting a threshold for interaction quality with PAEinter > 15 = "bad" interaction and PAEinter <15 = "good" interaction. To evaluate each mutation's impact on ORF2-ITGB1 binding, a score based on the ratio of good to bad predictions compared to wild-type interactions was

created:

$$\text{impact}(M) = \frac{2(GPI_{WT} - GPI_M)}{GPI_{MAX} + \text{sgn}(GPI_M - GPI_{WT})}$$

with GPI = number of Good Predicted Interactions of a given mutation ($GPI_M$). This score ranges from −1 to 1, with −1 indicating complete improvement in interaction prediction relative to the wild type ($GPI_{WT}$) (i.e., $GPI_{MAX}$) and 1 signifying complete degradation of binding predictions (i.e., zero good interactions). All scripts used and models produced in this study are available on Zenodo (DOI 10.5281/zenodo.14964884).

## Statistical analysis

Graphs and statistical analyses were performed using GraphPad PRISM 8. In all figures where p-values were calculated, the corresponding statistical test is listed in the figure legend. The exact p-values are listed in the source data file.

## Reporting summary

Further information on research design is available in the Nature Portfolio Reporting Summary linked to this article.

## Data availability

Proteomic data sets have been deposited to PRIDE with project accession: PXD052479. Access link: https://www.ebi.ac.uk/pride/review-dataset/be8b4508e9ba4790ad1894c4b79ec52f. Source data are provided with this paper.

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

## Acknowledgements

This project was supported by grants from the Deutsche Forschungsgemeinschaft (DFG, German Research Foundation) – Projektnummer – 272983813 SFB/TRR 179 and DA 1640/3-1 and from the German Center for Infection Research DZIF - TTU Hepatitis Project 05.823. VLDT was supported by the Chica and Heinz Schaller Foundation and JH was supported by a fellowship from the China Scholarship Council. E.S. was supported by the German Research Council (STE 1954/12-1 and 1954/14-1), German Centre for Infection Research (DZIF, TTU 05.823_00) and by the Ruhr University Bochum InnovationsFoRUM (Project: Host Microbe Interactions, IF-018N-22). U.K. was supported by the German Ministry of Education and Research (BMBF) within the LiSyM network [031L0042, 031L0048], the LiSyM-Cancer networks SMART-NAFLD [031L0256A] and C-TIP-HCC [031L0257C], MSCoreSys network SMART-CARE [031L0212B] and the German Center for Lung Research (DZL) [82DZL004A4] and funding by the Deutsche Forschungsgemeinschaft (DFG) within FerrOs [FOR5146]. TT was granted a postdoctoral fellowship by the ANRS-MIE (ANRS0548b and ANRS0541). For the publication fee we acknowledge financial support by Heidelberg University. The authors gratefully acknowledge Suzanne Emerson, Rainer Ulrich, Daniela Mauceri, and Britta Brügger for sharing reagents. We acknowledge Vibor Laketa, head of the Infectious Diseases Imaging Platform (IDIP) at the University Hospital Heidelberg for expert support. pSpCas9(BB)–2A-GFP (PX458) was a gift from Feng Zhang (Addgene plasmid # 48138; http://n2t.net/addgene:48138; RRID:Addgene 48138) and pWPI was a gift from Didier Trono (Addgene plasmid # 12254; http://n2t.net/addgene:12254; RRID:Addgene_12254). We thank Ann-Kathrin Mehnert for valuable and helpful discussions.

## Author contributions

Conceptualization, R.M.F, S.L. and V.L.D.T.; Methodology, R.M.F, Z.E.; Investigation, R.M.F, P.J., Z.E., J.A.W, J.M., H.C., S.B.D.L, B.H., M.K., J.H., A.F., T.T.; Resources: T.B., M.B., U.K., E.S., P.Y.L; Software, T.T., R.M.F.; Data analysis, R.M.F. and V.L.D.T.; Writing-original draft, R.M.F. and V.L.D.T.; Final draft, R.M.F., P.Y.L., S.L., and V.L.D.T.; Supervision, V.L.D.T.; Funding, V.L.D.T.

## Funding

## Competing interests

All authors declare no conflict of interest.
