## [Transparent Peer Review file · Nature Communications]

Integrin beta 1 facilitates non-enveloped hepatitis E virus cell entry through the recycling endosome

Corresponding Author: Dr Viet Loan Dao Thi

Version 0:

Reviewer comments:

Reviewer #1

(Remarks to the Author)

The study by Fu et al. describes a role for integrin $\beta 1$ in hepatitis E virus (HEV) entry. Using a novel RNA-FISH-based imaging assay, the authors show that integrin $\beta 1$ is important for cell attachment and internalization of naked HEV (nHEV) but not quasi-enveloped HEV (eHEV) particles. They provide evidence that a DGR motif in the HEV capsid mediates the interaction with integrins since a D552 E substitution in this motif reduced nHEV cell binding and infectivity. They further show that nHEV entry was dependent on Rab5 and Rab11, whereas eHEV entry depends on Rab5 and Rab7. Thus, the authors conclude that nHEV and eHEV enter the cell through Rab11+ recycling endosomes and late endosomes, respectively. Finally, they show that entry of both virion types requires endosomal acidification and lysosomal proteases. The study provides valuable new information about the poorly defined HEV entry process. The RNA-FISH-based imaging assay is quite novel and allows for the detailed study of early steps in HEV entry. The data were clearly presented and of high quality. However, the effect of integrin $\beta 1$ depletion on HEV entry appears to be rather moderate (two-fold), and more experimental evidence would be needed to demonstrate the role of recycling endosomes in nHEV entry. Finally, the requirements of endosomal acidification and proteases for HEV entry (Figs. 4-6) have been recently reported (PMID: 38728662), which diminishes the novelty of the study. Specific comments are below.

1. The authors show that integrin $\beta 1$ depletion reduced cell attachment of nHEV but not eHEV. This result is very different from a previous study from the Lemon group showing that while integrin $\beta 1$ depletion significantly reduced the entry of both naked and quasi-enveloped hepatitis A virus (HAV), there was no effect on cell binding of both virion types. While the isoDGR motif found in the HEV capsid may possibly explain the specificity for the nHEV, the different effects of integrin $\beta 1$ depletion on the entry of quasi-enveloped HAV and HEV remain unexplained. This is an important point that should be discussed.

2. HEV has a very high particle to FFU ratio. The authors used a rather low MOI (0.1 GE/cell) in some experiments, which presumably results in very few cells infected. Please provide the percentage of infected cells at this low MOI. Also in many figures, the FFU were normalized to the controls and shown as “%” of controls. It would be more informative to show the actual FFU numbers.

3. The authors show that Rab11-GFP colocalized with naked HEV and Rab11 knockdown impaired nHEV uncoating. It would be important to also show the effect of Rab11 knockdown on virus infectivity. As cargoes sorted into the recycling endosomes usually recycle back to the cell surface, it would be important to know how efficiently nHEV gains successful entry especially at a low MOI condition used in this study (MOI=0.1 GE/cell). In addition to Rab11, it would be important to also test the involvement of other components of the recycling endosomes (such as Rab4, EHD1 and MICAL-L1).

4. The idea that the isoDGR motif mediates the HEV capsid and integrin $\beta 1$ interaction is quite interesting. The authors show that HEV with a D522E mutation reduced cell binding and infectivity by 50%. However, more evidence would be needed to demonstrate this specific interaction. Is the RGD motif well exposed in the structure? It would be helpful to use molecular docking to show how integrin $\beta 1$ would possibly interact with the HEV capsid. Ideally, a direct interaction between the capsid or VLP (wildtype vs the D522E mutant) and integrin $\beta 1$ should be demonstrated. Also, does the cell binding and infectivity of the D522E mutant become insensitive to integrin $\beta 1$ depletion and RGD peptide inhibition, and does the D522E mutation affect eHEV cell binding and internalization?

Reviewer #2

(Remarks to the Author)

The study of Fu and colleagues focuses on mechanisms of hepatitis E virus (HEV) entry step into host cells, which is a poorly understood step of life cycle. Although HEV is found as a naked virus (nHEV) in bile and feces, it circulates as a quasi-enveloped virus (eHEV) in the bloodstream. Previously, it has been shown that nHEV and eHEV use different cellular factors and entry mechanisms to infect the host cells. In their study, the authors first found that Integrin beta1 (ITGB1) mediates nHEV cell entry, but not eHEV. They then used RNA-FISH to visualize and study the entry pathways of both nHEV and eHEV particles. They used their RNA-FISH approach and ITGB1 KO/rescued cells to validate the role of ITGB1 in nHEV entry. They also studied the interaction between the ORF2 capsid protein and ITGB1. Next, they carried out kinetic studies of HEV uncoating using a combination of RNA-FISH and capsid staining. Using lysosomotropic agents, silencing Rab proteins and colocalization studies, they demonstrated that ITGB1 is likely to drive nHEV entry, through Rab11-positive recycling endosomes. In contrast, eHEV particles use a classical endocytic pathway via Rab5a-positive early endosomes. Both nHEV and eHEV virions then converge on Rab7-positive late endosomes and lysosomes for uncoating by a cathepsin-dependent mechanism.

Overall, this study is interesting, well designed and provides important new insights in the HEV field. This study will be also interesting for a relatively broad audience of virologists. However, there are many issues that need to be addressed.

Major points:

-The role for ITGB1 in eHEV vs nHEV entry has only been demonstrated in S10-3 cells. eHEV control should be included for the other cell lines.

-In the text it is mentioned that throughout the study more genome equivalents per cell are used for eHEV than for nHEV to obtain comparable numbers for FFUs. What was the choice of choosing 0.1 GE/cell for nHEV and using 50x more for eHEV (5GE/cell) if supplemental fig. 2 shows a specific infectivity that increases only approximately 5x? In the following figures, e.g. the binding and internalization of Fig3A and 3B, less eHEV is used compared to nHEV to perform the assays. Why has this approach changed?

-Claims about the colocalization of cellular factors/ RNA must be confirmed by calculating Pearson's or Manders' coefficients on a significant number of cells (at least 30 cells, on single confocal slice), and not be based only on a section of a single cell (Fig. 3E, Fig. 5G-J, Fig. 6 A, F)

-Line 331 states that nHEV and eHEV preps are derived from density gradients, but for Fig. 1, nHEV and eHEV preps were also used. So are these also gradient-derived, or what exactly was used to perform the infection assays? This should be indicated in the figure legend.

-Sup Fig4: The figure legend indicates that HEV RNA in fractions was determined by RT-qPCR, however RNA levels are not shown in the figure. Please add RNA quantification on the graphs.

-Fig 2 & Fig 3: it is indicated that 8-9 fields from three independent experiments were used to calculate the number of HEV particles per cell. Please add the total number of cells analyzed to increase the robustness of the data.

-Lines 377-393: this part is particularly critical.

(i) "Confirming a direct interaction of the capsid with ITGB1, we detected a clear colocalisation of nHEV particles with ITGB1 on the cell membrane". What previous data show a direct interaction between capsid and ITGB1? Furthermore, the colocalization of 2 proteins (which here needs to be analyzed more robustly, see below) does not prove a direct interaction.

(ii) "we identified a reverse RGD "DGR" motif in the P-domain of the ORF2 sequence (Fig. 3F)" How this structure was generated and on the basis of which sequence, is not indicated in the mat&met.

(iii) " We also density-gradient purified both HEV389WT and HEV-D522E particles and found that the mutation did not impact progeny assembly (Suppl. Fig. 7). Surprisingly, gradients are not shown. Instead, the authors show results of percentage of RNA colocalized with capsid protein using an experimental approach on PEI-coated slides that is not described in the mat&met. Please show gradients of WT and HEV-D522E particles as in Sup Fig 4 (including RNAs). In addition, showing that D522E mutation does not affect eHEV infectivity (which is ITGB1-independent) will prove that the mutation does not induce ORF2 misfolding.

(iv) Fig 3G, please explain why WT & D522E RNA levels decrease over time instead of increasing?

(v) Direct capsid/ITGB1 interaction should be further addressed by proximity-ligation assay and/or pull-down assay. It could be interesting to then verify what happens with the HEV-D522E mutant as a control.

-Fig 4A, please describe how these experiments were performed. How probes can have access to encapsidated HEV RNA? What controls were used to assess assay specificity?

-Fig 4B, lower panel, what has happened to the HEV capsid, is it degraded or redistributed, can it be seen on wider fields?

-It is mentioned in line 443 that the data suggests that the cell entry of eHEV is slower and less productive than that of nHEV but can one really proof this by these data if different experimental procedures are performed for both of these HEV preps (different binding times and different amounts of GE/cell). Moreover, Fig. 4H does not show significant differences for both of the investigated preps.

-Fig. 5, regarding the use of EGFP-Rab proteins. How does the cellular localization of EGFP-Rab proteins correspond to

wild-type Rab proteins? These should be included as control.

-Fig 5E-F, Interestingly, the authors demonstrated that silencing of Rab11, but not Rab5a, resulted in a significant decrease of nHEV uncoating. In contrast, silencing of Rab5a, but not Rab11, resulted in a significant decrease of eHEV uncoating. Rab7 silencing affected both eHEV and nHEV uncoating. Since Yin et al observed contrasting effects of Rab5 and Rab7 silencing on nHEV and eHEV infection, the authors may consider adding nHEV and eHEV infection levels in siRab cells.

-Line 477, "Interestingly, we also observed ITGB1 in nHEV and Rab 11-positive endosomes, supporting our hypothesis that the interaction with ITGB1 sorts nHEV, but not eHEV particles, into Rab11-positive vesicles, with Rab11 being a hallmark of recycling endosomes ". To reach such a conclusion, the authors need to show ITGB1/RNA/Rab11 staining for eHEV. Fig5H-J left panels, it will be more informative to show ITGB1 staining instead of nuclei

-Discussion

The authors should contextualize their findings on ITGB1 involvement with those of Shrader et al showing modulation of HEV entry by EGFR. Did the authors check the impact of ITGB1 on EGFR expression levels ?

An additional section should be added to discuss in more detail the biological relevance of ITGB1 during viral infection, how is the expression of this protein on hepatocytes related to the in vivo situation and to naked particles? knowing that particles transfer from gut to liver via the portal vein and are therefore enveloped? What is then the significance of this receptor?

Minor points:

- Fig 1B and Fig 1F, why does the profile of ITGB1 protein differ between these two blots?

-Sup Fig1B, add intensity overlaps. Middle and low panels: scale bars are missing.

-Fig2B, please show in Sup Inf the images obtained with eHEV. Please also show images obtained with mock cells to assess the background of the green RNA probe.

-Fig2C, data should be presented as number of HEV particles/cell, as in Fig2B & Sup Fig5

-Fig 3E, please show data obtained with eHEV and KO cells.

-Fig 6G: please add a legend. The involvement of Rab5c is suggested but could one elaborate on this in more detail.

Typos:

-Figure 1A: PLC/PRF/5

-Figure 1D/y-axis: Relative nHEV FFU normalized to WT ? and not DMSO

-Figure 5/legend: (A) and (B) S10-3 cells were treatedof bafilomycin A (BFA), concanamycin A (ConA)....BFA is generally used as an abbreviation for Brefeldin A, please use BafA1 as an abbreviation for Bafilomycin A1

-line 196: ...immunofluorescence staining

-line 436, add reference of Yin et al

-lines 1125-1131, edit legend

Reviewer #3

(Remarks to the Author)

The manuscript submitted by Fu et al., delves into an important step of the HEV life cycle, which I believe will be of interest to researchers working on HEV. nHEV, which is transmitted through feco-oral route, remains a major public health concern and identification of the host factor involved in virus entry is a significant step towards development of specific therapeutics. However, following concerns need to be addressed to improve the clarity and conclusions, as mentioned below.

1. Representative figures of FFU assays need to be shown to ascertain reliability of the assay.
2. The authors rely on FFU assay to interpret all the data on virus infection. In order to rule out the biasness of antibody towards nHEV, a western blot of ORF2 may be added in addition.
3. With reference to line 349, "To ensure that viral particles packaged full-length HEV genomes", it will be more convincing to incubate the cells for longer period, costain the cells with anti-ORF2 antibody and RNA-probe to demonstrate that functional genome was present. Partial genome containing ORF1 and ORF2 fragments cannot be ruled out by FISH.
4. Refer to Fig 3H, the difference is not very striking. That should be explained.
5. Refer to Fig 4A, it is expected to get empty particles, but why only RNA is visible (atleast 2 dots)? That raises concern on technicality of the experiment.
6. Refer to Fig 4D, why BFA is not totally abrogating HEV RNA? Also, BFA use need to be mentioned in the text while writing the result for 4D.
7. Refer to Fig 4, at 24 hour time point, progeny virus could also account for the signal.
8. A recent study has shown that Rab11 positive endocytic recycling compartment serves as replication factory for HEV (doi: 10.1007/s00018-022-04646-y). If nHEV entry and assembly is dependent on Rab11, results need to be carefully interpreted. Authors need to explain their data accordingly. A replication defective mutant virus may help in distinguishing between entry

and progeny virus assembly steps.

9. Refer to Fig 4H, why is there a break in x axis? It seems not to affect the representation.

10. Refer to Fig 4H, is it possible to supplement the data with HEV RNA RT-PCR of membrane fraction?

Version 1:

Reviewer comments:

Reviewer #1

(Remarks to the Author)

The revised manuscript has been greatly improved in the clarity and completeness of several experiments. However, several issues remain to be addressed.

1. A major concern is that knockout of ITGB1 led to only a 2-fold reduction in nHEV cell binding and infectivity. Thus, ITGB1 doesn't appear to play a critical role in nHEV entry. This rather moderate effect diminishes the significance of the study.
2. The direct binding between ITGB1 and the viral capsid remains unconvincing. New data are provided to support direct binding using AlphaFold modeling. However, the model based on ORF2 dimer indicates the ITGB binding site on the virion is inaccessible and therefore conformational changes in the virion are likely needed. What could be the trigger for such conformational changes? The authors should use HEV-LP to demonstrate direct binding between the capsid and ITGB1 and the role of the DGR motif using various DGR mutants.
3. It remains unclear how nHEV moves from the recycling endosomes to the lysosome for uncoating. Additional experimental evidence would be needed, as this seems to be an important aspect of the working model shown in Fig. 7. Also, the role of Rab5c in nHEV entry should be addressed to strengthen the model.

Reviewer #2

(Remarks to the Author)

Compared to the previous version, Fu and colleagues have significantly improved their manuscript. They have carefully and convincingly addressed my concerns, and implemented corresponding changes in the manuscript. Therefore, I believe that their manuscript is suitable for publication in Nature Communications.

Best regards,

Minor issues / typos.

Sup Fig 3B: Analyses were performed in S10-3 or S10-3-ITGA3 ? please clarify

Leg Sup Fig 3, line 68: Line graphs on the right show the fluorescence intensities of ITG1 and ITGA2-3-5 measured across the region of interest

Sup Fig 12, panels A, C and E: please increase size and resolution of these panels

p11 line 458: EPO - electroporation

p12 line 494: DRG > DGR

p14 line 604: PLC/PRE/5 > PLC/PRF/5

Fig. 4F: is not mentioned in the text

Reviewer #3

(Remarks to the Author)

The revised manuscript addresses all my queries and I am satisfied with the response of the authors. I recommend publication of this manuscript.

Point-by-point reply to the reviewers' comments

Reviewer #1 (Remarks to the Author):

The study by Fu et al. describes a role for integrin 1 in hepatitis E virus (HEV) entry. Using a novel RNA-FISH-based imaging assay, the authors show that integrin 1 is important for cell attachment and internalization of naked HEV (nHEV) but not quasi-enveloped HEV (eHEV) particles. They provide evidence that a DGR motif in the HEV capsid mediates the interaction with integrins since a D552 E substitution in this motif reduced nHEV cell binding and infectivity. They further show that nHEV entry was dependent on Rab7 and Rab11, whereas eHEV entry depends on Rab5 and Rab7. Thus, the authors conclude that nHEV and eHEV enter the cell through Rab11+ recycling endosomes and late endosomes, respectively. Finally, they show that entry of both virion types requires endosomal acidification and lysosomal proteases.

The study provides valuable new information about the poorly defined HEV entry process. The RNA-FISH-based imaging assay is quite novel and allows for the detailed study of early steps in HEV entry. The data were clearly presented and of high quality. However, the effect of integrin 1 depletion on HEV entry appears to be rather moderate (two-fold), and more experimental evidence would be needed to demonstrate the role of recycling endosomes in nHEV entry. Finally, the requirements of endosomal acidification and proteases for HEV entry (Figs. 4-6) have been recently reported (PMID: 38728662), which diminishes the novelty of the study.

We thank the reviewer for their positive evaluation of our study. Their helpful and constructive comments have allowed us to improve the manuscript.

Specific comments are below.

1. The authors show that integrin 1 depletion reduced cell attachment of nHEV but not eHEV. This result is very different from a previous study from the Lemon group showing that while integrin 1 depletion significantly reduced the entry of both naked and quasi-enveloped hepatitis A virus (HAV), there was no effect on cell binding of both virion types. While the isoDGR motif found in the HEV capsid may possibly explain the specificity for the nHEV, the different effects of integrin 1 depletion on the entry of quasi-enveloped HAV and HEV remain unexplained. This is an important point that should be discussed.

We thank the reviewer for this insightful comment. Indeed, due to the direct interaction mediated by the DGR motif in the HEV capsid, the impact of integrin depletion is likely to be very different from that on HAV entry.

Both eHAV and eHEV particles acquire their quasi-envelope by budding into MVBs, and at least one study has shown that ITGB1 heterodimers may be involved in extracellular vesicle (EV) uptake (Hussain et al., 2024, Int J Mol Sci). Therefore, we agree that ITGB1 could theoretically play a direct role in the uptake of quasi-enveloped HEV particles, even in the absence of a direct capsid interaction. One reason for the observed different dependencies could be the different cell types used in the two studies: While Rivera-Serrano and colleagues used the cervical cancer cell line HeLa to demonstrate the importance of ITGB1 for both naked and quasi-enveloped HAV particles, we mainly used a panel of hepatoma cell lines in our study. At least two studies have reported that ITGB3 and not B1 is critical for EV uptake in the breast cancer cell line MDA.MB.231 (Fuentes et al., 2020, Nat Commun, Altei et al., 2020, Cell Commun Signal). Therefore, EV and/or quasi-enveloped particle uptake may be mediated differently in hepatoma cells than in cervical cancer cells. Consistent with this, the dependence of eHAV particles on the TIM-1 receptor

appears to be cell type dependent, as a significant decrease in eHAV was previously observed in TIM1-KO Vero cells, but not in TIM1-KO Huh-7.5 cells (Das et al., 2017, mBio).

We agree with the Reviewer that this is overall an important aspect that should be further investigated in the future and we have added it to the Discussion section of the revised manuscript (lines 712-724).

2. HEV has a very high particle to FFU ratio. The authors used a rather low MOI (0.1 GE/cell) in some experiments, which presumably results in very few cells infected. Please provide the percentage of infected cells at this low MOI. Also in many figures, the FFU were normalized to the controls and shown as “%” of controls. It would be more informative to show the actual FFU numbers.

Throughout our manuscript, we aimed to analyse, for example, the effect of ITGB1 deficiency on HEV infection in a quantitative manner. In this respect, we assessed focus-forming units per ml (FFU/ml), as these, together with plaque-forming units, are the most established, accurate and reliable methods for quantifying viral titers and/or viral infection events (Baer & Kenn-Hall, 2014, J Vis Exp). In addition, and in agreement with the reviewers' statement, we have found that HEV has low specific infectivity (i.e. infectious particles per physical particles assessed by quantifying viral genomes, Suppl. Fig. 5), which limits the large-scale production of high-titer HEV preparations. Finally, we have found that HEV infection spreads by cell division in cancer cells (data not published). For all these reasons, we aimed to quantify the number of FFUs rather than the percentage of infected cells.

However, to reliably assess the number of FFUs, well separated and countable foci are required. To achieve this, we inoculated the cells with a highly diluted viral suspension (at a MOI = 0.1 GE/cell), resulting in a low percentage of infected cells (0,17%, Suppl. Fig. 2). Therefore, we agree with the reviewer that at such a low MOI, only a small percentage of cells are infected. However, we have carefully chosen the MOI to yield quantifiable foci within the linear range of our assay, as too many infection events will simply lead to a plateau in countable foci.

We then recalculated the FFU numbers per ml based on our foci counts and volumes of virus prep used for infection and show these in the new Suppl. Fig. 3. We better explain the application of the low MOI in our infection assays and how the HEV infectivity rates were obtained in the Results section of the revised manuscript (lines 310-314). In addition, in the new Suppl. Fig. 3, we provide non-normalised FFU/ml values for all infection data from the main figures, namely Figs. 1D, G, I&J, 3G, 5 A&B, 7B&C.

3. The authors show that Rab11-GFP colocalized with naked HEV and Rab11 knockdown impaired nHEV uncoating. It would be important to also show the effect of Rab11 knockdown on virus infectivity. As cargoes sorted into the recycling endosomes usually recycle back to the cell surface, it would be important to know how efficiently nHEV gains successful entry especially at a low MOI condition used in this study (MOI=0.1 GE/cell).

We appreciate the reviewer's insightful comment. Indeed, our data show that both eHEV and nHEV entry is relatively inefficient: infection of cells with gradient-purified nHEV particles, at an MOI of 0.1 GE/cell (which is equivalent to approximately 3000 infectious particles, as titered in end-point dilution assays on S10-3 cells), yields only ~100-200 foci per well. This is consistent with the low specific infectivity of e/nHEV particles (Suppl. Fig. 5) as mentioned in our response

above. As shown in Fig. 2B, even at a high MOI of 20 GE/cell, only half of the bound nHEV particles are internalised. This may be due to trafficking of particles back to the cell surface via the recycling endosome.

We have attempted to assess the effect of Rab11 downregulation on e/nHEV infection. Unfortunately, as explained in our response to Reviewer 2, we were unable to replicate previously published results by Yin and colleagues (2014, JVI). We speculate that downregulation of adaptor proteins along the endolysosomal pathway may also be critical for HEV genome replication and progeny assembly. In addition, it may also affect innate immune responses. Please see our response to Reviewer 2 (# 14) for corresponding results and further details.

In addition to Rab11, it would be important to also test the involvement of other components of the recycling endosomes (such as Rab4, EHD1 and MICAL-L1).

Thank you for this helpful suggestion. Accordingly, we performed our RNA-FISH-based entry assay to investigate the effect of other recycling endosome components on nHEV uncoating using respective siRNAs. We found that knockdown of components EHD1 and MICAL-L1 indeed resulted in an increase in the percentage of uncoated RNA compared to the non-targeted control. These results further confirm the involvement of the recycling endosome in nHEV entry and we show them in the new Suppl. Fig. 16. We have added this information to the Results section of the revised manuscript (lines 561-564).

Interestingly, we did not observe a significant effect upon Rab4 downregulation. According to a study by Wilson and colleagues (2023, J Cell Sci), Rab4 and Rab11 regulate distinct recycling pathways, suggesting a specific role for Rab11 in directing nHEV entry through the recycling endosome.

4. The idea that the isoDGR motif mediates the HEV capsid and integrin 1 interaction is quite interesting. The authors show that HEV with a D522E mutation reduced cell binding and infectivity by 50%. However, more evidence would be needed to demonstrate this specific interaction. Is the RGD motif well exposed in the structure?

The DGR motif is indeed located in the well-exposed protrusion domain of the HEV capsid, as shown in Suppl. Fig. 10A (previous Fig. 3F). We have added this important detail to the manuscript (line 446). Structurally, this region has been proposed to be involved in receptor binding and antibody neutralisation (Guu et al., 2009, PNAS). To further support the hypothesis that the HEV capsid interacts with ITGB1 heterodimers through the RGD motif, we performed additional experiments as requested below:

4a) It would be helpful to use molecular docking to show how integrin 1 would possibly interact with the HEV capsid.

We collaborated with Dr. Thibault Tubiana, who used AlphaFold2 to model the interaction between ITGB1 and the DGR motif (new Fig. 3I). Various models were generated using both monomeric and dimeric forms of the ORF2 P domain, with both full-length and truncated forms of ITGB1, to identify a high-scoring binding interface with low Predicted Aligned Error (PAE) (new Suppl. Fig. 12).

Based on the models exhibiting low PAE, we identified critical amino acid residues at the ORF2P-ITGB1 interface for alanine mutagenesis using AlphaFold. These results strongly support our

hypothesis that the DGR motif acts as a binding motif. In total, over 300 AlphaFold models in the course of this analysis were generated and we have added these new results to the Result section (lines 469-491).

4b) Ideally, a direct interaction between the capsid or VLP (wildtype vs the D522E mutant) and integrin 1 should be demonstrated.

Thank you for encouraging us to strengthen this aspect. We have performed a proximity ligation assay (PLA) to confirm the colocalisation between ITGB1 and HEV particles on the cell membrane (new Fig. 3J). Critically, we found that the D522E mutant interacted less with ITGB1 as compared to the WT, thereby confirming the potential interaction via the DGR motif in the capsid. We have added these results together with the modelling data to our revised Results section of the manuscript (lines 492-495).

4c) Also, does the cell binding and infectivity of the D522E mutant become insensitive to integrin 1 depletion and RGD peptide inhibition,

We used siRNA to knockdown ITGB1 expression and conducted binding and infectivity assays using wild-type (WT) and D522E mutant nHEV. Our results indicate that while D522E-nHEV was not entirely insensitive to ITGB1 depletion, its sensitivity was reduced compared to WT-nHEV. A similar trend was observed during D522E-nHEV infection in the presence of the RGD peptide, where the peptide exhibited a weaker inhibitory effect on the mutant virus compared to WT-nHEV (Suppl. Fig. 11). These findings align with the expectation that the D522E mutation, involving a conservative substitution (D to E), is relatively mild. Correspondingly, the reduction in viral binding and infectivity caused by this mutation was also modest (Fig. 3G and 3H). We have added these findings to the Results section of our revised manuscript (lines 464-468).

To avoid affecting viral replication, we deliberately chose a mild mutation rather than a more disruptive change. Our attempts to introduce a high impact mutation by replacing the R residue with an A resulted in a strong decrease in viral replication, as compared to the WT virus (Revision Fig. 1A). This harsh mutation also significantly altered the cellular distribution of ORF2 5 days post-electroporation (Revision Fig. 1B), further suggesting a disruption of viral replication and/or assembly. In contrast, the D522E mutant behaved similarly to the WT virus. We have therefore chosen the D522E mutation as a model to study the effect of the DGR motif on capsid interaction with ITGB1 and have added this explanation to the Results section of the revised manuscript (lines 455-456).

Revision Figure 1. Introduction of a harsh mutation into the ORF2 DGR motif impacts viral replication and ORF2 expression. (A) Replication levels of the WT, D522E and R524A virus were determined by electroporating S10-3 cells with HEV RNA and harvesting cells 1-, 3-, 5-, and 7-days post-electroporation. HEV RNA copies were determined by qRT-PCR. The replication-incompetent GNN mutant serves as a negative control. $n = 4$ replicates from two independent experiments. (B) S10-3 cells were electroporated with HEV RNA and cells were fixed 7 days post-electroporation. ORF2 expression was visualized by IF using an antibody against ORF2 (magenta) and counterstain using DAPI (blue). The images were taken on a CellDiscoverer 7 microscope. Scale bar = 20 μm

4d) and does the D522E mutation affect eHEV cell binding and internalization?

We have tested the capability of enveloped D522E particles to bind and be internalised by cells. As shown in the new Suppl. Fig. 10E, D522E eHEV particles have an even greater ability to bind to and be internalised by the cell than WT eHEV particles. This is consistent with our observation that they are more infectious (see response to Reviewer 2 #8(iii) and new Suppl. Fig. 10D). We can therefore rule out any deleterious effect of the mutation on capsid assembly and conclude that the mutation does indeed affect the infectivity of nHEV particles, probably by affecting the interaction between the capsid and ITGB1 (lines 464-468).

Reviewer #2 (Remarks to the Author):

The study of Fu and colleagues focuses on mechanisms of hepatitis E virus (HEV) entry step into host cells, which is a poorly understood step of life cycle. Although HEV is found as a naked virus (nHEV) in bile and feces, it circulates as a quasi-enveloped virus (eHEV) in the bloodstream. Previously, it has been shown that nHEV and eHEV use different cellular factors and entry mechanisms to infect the host cells. In their study, the authors first found that Integrin beta1 (ITGB1) mediates nHEV cell entry, but not eHEV. They then used RNA-FISH to visualize and study the entry pathways of both nHEV and eHEV particles. They used their RNA-FISH approach and ITGB1 KO/rescued cells to validate the role of ITGB1 in nHEV entry. They also studied the interaction between the ORF2 capsid protein and ITGB1. Next, they carried out kinetic studies of HEV uncoating using a combination of RNA-FISH and capsid staining. Using lysosomotropic

agents, silencing Rab proteins and colocalization studies, they demonstrated that ITGB1 is likely to drive nHEV entry, through Rab11-positive recycling endosomes. In contrast, eHEV particles use a classical endocytic pathway via Rab5a-positive early endosomes. Both nHEV and eHEV virions then converge on Rab7-positive late endosomes and lysosomes for uncoating by a cathepsin-dependent mechanism.

Overall, this study is interesting, well designed and provides important new insights in the HEV field. This study will be also interesting for a relatively broad audience of virologists. However, there are many issues that need to be addressed.

We thank the reviewer for their positive comments on our study and their helpful and constructive suggestions, which allowed us to improve the manuscript.

Major points:

1. The role for ITGB1 in eHEV vs nHEV entry has only been demonstrated in S10-3 cells. eHEV control should be included for the other cell lines.

We thank the reviewer for raising this point. Accordingly, we used siRNA to downregulate ITGB1 in four additional cell lines and infected them with eHEV and nHEV. As shown in new Suppl. Fig. 6, depletion of ITGB1 significantly reduced nHEV but not eHEV infection, in all cell lines tested, similar to our observation in S10-3 cells (lines 358-362).

2. In the text it is mentioned that throughout the study more genome equivalents per cell are used for eHEV than for nHEV to obtain comparable numbers for FFUs. What was the choice of choosing 0.1 GE/cell for nHEV and using 50x more for eHEV (5GE/cell) if supplemental fig. 2 shows a specific infectivity that increases only approximately 5x?

We have updated the corresponding Figure (Suppl. Fig. 5) showing the specific infectivities of nHEV and eHEV and now express them as "genome equivalents per one FFU" (GE: FFU ratio) for easier comparison. The specific infectivity of nHEV is approximately 100-fold higher than that of eHEV particles, which is in agreement with our previous study (Dao Thi et al.; 2020, Nat Commun, also in Suppl. Fig. 5). To best account for this difference, we used 50 times more eHEV particles per cell (5 GE/cell for eHEV vs. 0.1 GE/cell for nHEV) to achieve comparable FFU counts. As the volume of eHEV particles that we can add to a well for our infection assays remains a limiting factor, we could not match the 100x perfectly. We have added this explanation to the results section (lines 343-350).

3. In the following figures, e.g. the binding and internalization of Fig3A and 3B, less eHEV is used compared to nHEV to perform the assays. Why has this approach changed?

We harvest nHEV particles from cell lysates, whereas eHEV particles are harvested from cell supernatants (similar to others and our previous study Dao Thi et al.; 2020, Nat Commun). Therefore, the volumes and the corresponding concentrations (virus GE/ml) that can be achieved are generally very different. In addition to the higher volume, we cannot concentrate eHEV particles by sucrose cushion to the same extent as nHEV particles (harvested in PBS) due to the presence of FBS in the culture medium. Consequently, the copy numbers per volume/ml are significantly lower for eHEV than for nHEV.

For the binding assay, a high number of viral particles must be added to achieve a quantifiable number of bound particles per cell and allow significant statistical analyses. However, the volume and therefore maximum number of eHEV particles that can be added is limited by the culture well size and the degree of medium dilution. Adding only purified virus prep to the cell in the absence of culture medium is detrimental to cell health.

For the infection assays on the other hand, we generally applied a low MOI. This allowed us to better adjust the volumes of eHEV and nHEV in order to reach comparable FFU counts without compromising cell viability. We hope this clarifies the different MOIs applied in the different assays used and have added these explanations to the Result section of the revised manuscript (lines 343-350, lines 393-396).

Unfortunately, all studies that aim to compare these two forms of the HEV particles have to deal with these differences (harvesting methods, maximum concentrations that can be achieved, differential specific infectivities, etc.). As we understand the limitations of directly comparing, for example their kinetics during cell entry, we have removed this aspect from the manuscript. As for the other aspects of this study, we sincerely believe that the way in which we present our results is the best possible way of taking these differences into account.

4. Claims about the colocalization of cellular factors/ RNA must be confirmed by calculating Pearson's or Manders' coefficients on a significant number of cells (at least 30 cells, on single confocal slice), and not be based only on a section of a single cell (Fig. 3E, Fig. 5G-J, Fig. 6 A, F). We thank the Reviewer for this constructive comment. We have re-analysed cells for each colocalisation image and calculated the Mander's correlation coefficient (Dunn et al., 2011, Am J Physiol Cell Physiol). This analysis shows that early after binding, nHEV RNA colocalises more with Rab11, while eHEV RNA colocalises more with Rab5-positive vesicles. At later time points, both genomes of both particle types colocalise with Rab7-positive vesicles. We have added the respective graphs to the new Fig. 3E, Fig. 6E-G, and Fig. 7A, G of the revised manuscript.

5. Line 331 states that nHEV and eHEV preps are derived from density gradients, but for Fig. 1, nHEV and eHEV preps were also used. So are these also gradient-derived, or what exactly was used to perform the infection assays? This should be indicated in the figure legend.

Yes, this is correct and we apologise for the error. We have used gradient purified virus preparations throughout the manuscript. We now mention the use of density-purified nHEV particles when describing the results of Fig. 1 and have added this information to Figure legend 1 (line 1032). In addition, we have added a sentence clarifying this in the Material and Methods section of the revised manuscript (lines 130-131).

6. Sup Fig4: The figure legend indicates that HEV RNA in fractions was determined by RT-qPCR, however RNA levels are not shown in the figure. Please add RNA quantification on the graphs. We apologise for this oversight and have added the HEV RNA quantification to the revised Suppl. Fig. 1.

7. Fig 2 & Fig 3: it is indicated that 8-9 fields from three independent experiments were used to calculate the number of HEV particles per cell. Please add the total number of cells analyzed to increase the robustness of the data

We have added the total number of cells analysed for each panel in Figs. 2 and 3 to the respective figure legends (Lines 1064, 1068, 1071, 1084, 1088, 1093, 1111, 1119).

8. Lines 377-393: this part is particularly critical.

8(i) "Confirming a direct interaction of the capsid with ITGB1, we detected a clear colocalisation of nHEV particles with ITGB1 on the cell membrane". What previous data show a direct interaction between capsid and ITGB1? Furthermore, the colocalization of 2 proteins (which here needs to be analyzed more robustly, see below) does not prove a direct interaction.

We apologise for the wording and have softened this statement in the revised manuscript. However, to strengthen our hypothesis that the HEV capsid potentially interacts with ITGB1 heterodimers via its DGR motif, we have used Alphafold2 to model the interaction. In addition, we have performed a proximity ligation assay (new Figs. 3I & J). We have also included a colocalisation analysis of eHEV particles (by detecting the RNA) and ITGB1 as a control (Suppl. Fig. 9A), and calculated the Mander's coefficient for the colocalisation of nHEV or eHEV RNA/particles with ITGB1 to provide further confidence on their proximity (Suppl. Fig. 9B).

8(ii) "we identified a reverse RGD "DGR" motif in the P-domain of the ORF2 sequence (Fig. 3F)" How this structure was generated and on the basis of which sequence, is not indicated in the mat&met.

We apologise for not providing this information earlier. For this analysis, the protein structure of the HEV GT3 2712 strain was used (PDB ID: 2ZTN; Yamashita et al.; 2009, PNAS) and was visualized using PyMOL (version 3.0, Schrödinger, LLC). We have moved the original Fig. 3F to Supplemental Materials (Suppl. Fig. 10A) and updated the Supplementary Methods, together with information on the software used (lines 25-32). We have also added this information to the figure legend (line 148).

8(iii) " We also density-gradient purified both HEV389WT and HEV-D522E particles and found that the mutation did not impact progeny assembly (Suppl. Fig. 7). Surprisingly, gradients are not shown. Instead, the authors show results of percentage of RNA colocalized with capsid protein using an experimental approach on PEI-coated slides that is not described in the mat&met. Please show gradients of WT and HEV-D522E particles as in Sup Fig 4 (including RNAs).

These are very constructive points. Accordingly, we have updated the Materials and Methods section to describe the *ex vivo* RNAscope analysis of HEV particles immobilised on PEI-coated slides (lines 261-267). We have performed the gradient density separation analysis of WT-HEV and D522E-HEV particles and added the results to the revised Supplementary Material (New Suppl. Fig. 10B, referred to in line 460). We have found that the distribution of infectious particles and viral RNA of the D522E mutant was similar to that of the WT virus, for both nHEV and eHEV particles.

In addition, showing that D522E mutation does not affect eHEV infectivity (which is ITGB1-independent) will prove that the mutation does not induce ORF2 misfolding.

We inoculated S10-3 cells with equal copy numbers of WT- and D522E-eHEV and found that the mutation did not reduce eHEV infectivity, but rather increased it by a small, but significant amount. This observation allows us to rule out that the mutation affects progeny assembly and/or ORF2

misfolding. We have included this result in the new Suppl. Fig. 10D, E and describe it in the Results section of our revised manuscript (lines 464-468).

8 (iv) Fig 3G, please explain why WT & D522E RNA levels decrease over time instead of increasing?

To compare genome replication levels between WT-HEV and D522E-HEV independently of progeny particle assembly and cell entry, we delivered the infectious genomes into the cells by electroporation. Since we electroporate the cells with a relatively high number of genomes, the RNA levels appear to decrease over time. However, the GNN mutant, in which we have mutated the catalytic site of the RdRp, shows that both WT-HEV and D522E-HEV replicate actively and to the same extent. We have added an explanation to the results section in the revised manuscript (lines 457-459).

8 (v) Direct capsid/ITGB1 interaction should be further addressed by proximity-ligation assay and/or pull-down assay. It could be interesting to then verify what happens with the HEV-D522E mutant as a control.

Also in response to Reviewer 1 (#4b), we have performed a proximity ligation assay showing a proximal interaction between ITGB1 and the HEV capsid on cells that was impaired by mutation of the DGR motif. We have included these results in the new Fig. 3J of the revised manuscript and thank the Reviewers for encouraging us to perform this important experiment. We also added these results to the Results section of the revised manuscript (lines 492-495).

9. Fig 4A, please describe how these experiments were performed. How probes can have access to encapsidated HEV RNA?

We apologise for the lack of explanation for this figure. According to the study by Ma and colleagues (2020, Mol Ther Methods Clin Dev), treatment with Triton-X 100 can lead to disintegration of non-enveloped and enterically transmitted Adenovirus capsids. In our study, we used Triton-X 100 (0.1% v/v in PBS) to permeabilise the HEV capsid prior to probe hybridisation. We have added this information to the new description of the *ex vivo* RNAscope analysis of HEV particles in the Material and Methods section (lines 264-266).

10. What controls were used to assess assay specificity?

To assess the specificity of the assay, we performed a panel of additional assays: We used the ORF1 probe to hybridise mock-infected cells, which gave no detectable signal (new Suppl. Fig. 7A). We also stained immobilised *ex vivo* cell-free HEV particles with a control probe targeting cellular GAPDH RNA (new Suppl. Fig. 7B). While the GAPDH probe effectively detected RNA in S10-3 cells (new Suppl. Fig. 7C), no signal was observed in *ex vivo* HEV particles. We now mention these important controls in our manuscript (lines 403-406).

11. Fig 4B, lower panel, what has happened to the HEV capsid, is it degraded or redistributed, can it be seen on wider fields?

The study by Kloehn and colleagues (2024, Hepatology) demonstrated that the HEV capsid undergoes processing by lysosomal cathepsins during viral entry, with cleavage products observed when the capsid is digested with cathepsin L. In our manuscript, we also found that

eHEV and nHEV entry require cathepsin activity (New Fig. 7). Based on these findings, we hypothesise that the capsid is degraded within the lysosome. The detection of single ORF2 proteins compared to an assembled capsid containing an estimated 180 copies of ORF2 protein (Xing et al.; 2010, JBC) is likely below the detection limit of the microscopy used. As shown in Revision Fig. 2, which shows a wider field of the image shown in Fig. 4B, occasionally, some faint capsid signals (as indicated by the white arrows) can be still detected which may correspond to capsid fragments. These signals can be also seen in the original Fig. 4B. We therefore mention this also in our revised manuscript (lines 589-591).

Revision Figure 2. Detection of faint capsid signals likely corresponds to partially degraded capsids. S10-3 cells were inoculated with nHEV (MOI = 30 GE/cell) and incubated for 2 h at 4 °C to allow binding followed by inoculum removal and 24 h at 37°C for internalisation. After fixation, cells were stained for HEV capsids (magenta) and HEV genomes (green) detected by immunofluorescence staining and RNA-FISH (version 1 kit) using the ORF1 probe, respectively. Faint capsid signals not colocalising with HEV RNA are marked with arrows. Scale bar = 10 µm.

12. It is mentioned in line 443 that the data suggests that the cell entry of eHEV is slower and less productive than that of nHEV but can one really proof this by these data if different experimental procedures are performed for both of these HEV preps (different binding times and different amounts of GE/cell). Moreover, Fig. 4H does not show significant differences for both of the investigated preps.

We agree with the reviewer that a series of comparative studies is needed to make the claim that eHEV cell entry is slower and less productive than nHEV cell entry. As we continue to be limited by the low titers imposed by the low specific infectivity of eHEV particles, and due to space constraints within the amount of data already presented, we have decided to remove Figs. 4G and H together with this claim from the Results and Discussion section. We believe that our study is not compromised by the absence of this claim due to important and novel aspects such as the use of RNA-FISH as an imaging approach to study nHEV and eHEV cell entry and the role of integrins and endocytic acidification along this process.

13. Fig. 5, regarding the use of EGFP-Rab proteins. How does the cellular localization of EGFP-Rab proteins correspond to wild-type Rab proteins? These should be included as control.

This is an interesting point. In the present study, we used EGFP-Rab5, 7 and 11 and EGFP-LAMP1 constructs that have been previously published and validated (Lozach et al., 2010, Cell Host Microbe). To confirm that the GFP signals could be used for our colocalisation analysis with HEV particles in Fig. 6, we co-stained EGFP-Rab5, 7 and 11 and EGFP-LAMP1 proteins with the

corresponding antibodies. As shown in Revision Fig. 3, we found that both endogenous and ectopic proteins had a similar subcellular localisation and that they colocalised strongly. The antibodies certainly also stained the endogenous proteins (magenta) that were not positive for the GFP signal, while the majority of the GFP signal colocalised with the respective antibody signal.

Revision Figure 3. Colocalisation of EGFP-tagged adaptor molecules with specific antibodies. S10-3 cells ectopically expressing EGFP-tagged endosomal markers were fixed and co-stained with antibodies against the protein of interest (magenta) and counterstained with DAPI (blue). The images were taken on a Zeiss Airyscan LSM900 confocal microscope and shown are maximum projections of single confocal slices. Scale bar = 10 μ m.

14. Fig 5E-F, Interestingly, the authors demonstrated that silencing of Rab11, but not Rab5a, resulted in a significant decrease of nHEV uncoating. In contrast, silencing of Rab5a, but not Rab11, resulted in a significant decrease of eHEV uncoating. Rab7 silencing affected both eHEV and nHEV uncoating. Since Yin et al observed contrasting effects of Rab5 and Rab7 silencing on nHEV and eHEV infection, the authors may consider adding nHEV and eHEV infection levels in siRab cells.

Unfortunately, and despite testing multiple conditions, we cannot reproduce the data of Yin et al. (2016, JVI). We have downregulated Rab5, 7 and 11, infected the cells at low and high MOIs in both S10-3 and HepG3/C3A cells, and assessed FFUs early or late after infection (see Revision Fig. 4, see downregulation efficiency Suppl. Fig. 15). Curiously, rather than reducing infection, downregulation of Rab proteins often enhanced HEV infection without a clear pattern or trend.

We can only speculate about the underlying causes: Several studies have meanwhile shown that the endolysosomal system is likely involved in HEV genome replication and/or progeny assembly and release (e.g.: Oechslin et al., 2023, PNAS; Glitscher et al.; 2024, Cell Mol Gastroenterol Hepatol). Professor Cocquerel's group named these endolysosomal- and more specifically, endocytic recycling-derived compartments "viral factories" (Bentaleb et al.; 2022, Cell Mol Life Sci). Based on these observations, we speculate that the downregulation of Rab family proteins not only affects HEV entry, but possibly also replication and progeny release. A defect in progeny release could lead to their accumulation within infected cells and therefore to the detection of FFUs that would otherwise fall below the sensitivity threshold of our analysis. This could lead to an increase in FFU counts while decreasing some viral entry events. In addition, endolysosomal trafficking is critical for innate immune responses (e.g. Guichard et al.; 2014, Nat Rev Microbiol) and we have recently found that replication- and infection-limiting cell-intrinsic responses are induced upon HEV infection at later stages of infection (Mehnert et al., 2025, bioRxiv). Taken together, these observations compromise the generation of fully conclusive results on cell entry from end-point infection assays and highlight the need for an assay that allows the separation of cell entry from the rest of the viral life cycle.

Revision Figure 4. Downregulation of Rab proteins has differential impacts on HEV infection. (A) and (B) S10-3 cells were transfected with 100 nM on-target pool siRNAs directed against Rab5, Rab7, Rab11 and a NT-control. 48 h post-transfection, the cells were inoculated with nHEV (MOI = 0.1 GE/cell) or eHEV (MOI = 5 GE/cell) and incubated for 24 h at 37°C. Virus was removed after 24 h and HEV infection was quantified by counting ORF2-positive FFUs (A) 5 days or (B) 48 h post-infection. Images of infected wells were taken with a Zeiss CellDiscoverer 7 microscope and the number of FFU was counted manually.

n = 2-9 replicates from 1-2 independent experiments. (A) Statistical comparison was performed by one-way ANOVA. ***: $p < 0.001$; ****: $p < 0.0001$; ns, non-significant (C) HepG2/C3A cells were transfected with 100 nM on-target pool siRNAs directed against Rab5, Rab7, Rab11 and a NT-control. 48 h post-transfection, the cells were inoculated with nHEV (MOI = 0.1 GE/cell) or eHEV (MOI = 5 GE/cell) and incubated for 24 h at 37°C. Virus was removed after 24 h and HEV infection was quantified by counting ORF2-positive FFUs 5 days post-infection. Images of entire infected wells were taken with a Zeiss CellDiscoverer 7 microscope and the number of FFU was counted manually. n = 2 replicates from 1 experiment. (D) HepG2/C3A cells were transfected with 100 nM on-target pool siRNAs directed against Rab5, Rab7, Rab11 and a NT-control. 48 h post-transfection, the cells were inoculated with nHEV (MOI = 30 GE/cell) or eHEV (MOI = 20 GE/cell) and incubated for 24 h at 37°C. Virus was removed after 24 h and HEV infection was quantified by counting the percentage of infected cells 5 days post-infection. n = 1-2 replicates from 1-2 experiments.

15. Line 477, "Interestingly, we also observed ITGB1 in nHEV and Rab 11-positive endosomes, supporting our hypothesis that the interaction with ITGB1 sorts nHEV, but not eHEV particles, into Rab11-positive vesicles, with Rab11 being a hallmark of recycling endosomes ". To reach such a conclusion, the authors need to show ITGB1/RNA/Rab11 staining for eHEV.

We agree with this reviewer and therefore added ITGB1 staining to each panel in Fig.5 G-J in the revised manuscript. As shown in the new images, RNA of nHEV but not eHEV particles (Fig. 3F) colocalised with ITGB1 and Rab11.

16. Fig5H-J left panels, it will be more informative to show ITGB1 staining instead of nuclei

Thank you, in addition to the nuclear staining, we have added panels of ITGB1 staining throughout the new Fig. 6E-G (previous Fig. 5H-J). We have also replaced the line profiles with the Mander's overlap coefficient.

17. Discussion

The authors should contextualize their findings on ITGB1 involvement with those of Shrader et al showing modulation of HEV entry by EGFR. Did the authors check the impact of ITGB1 on EGFR expression levels?

This is an interesting point. Indeed, both $\beta 1$ and $\beta 3$ integrins have been shown to associate with EGFR, activate EGFR in a ligand-independent manner (e.g. activate EGFR signaling through integrin binding, Wang et al.; 2003, JBC) and synergistically enhance EGFR signaling (Rubio et al.; 2023, Theranostics) and several viruses have been reported to exploit EGFR and integrins simultaneously in their life cycle (e.g. HMCV: Lee et al.; 2021, Front Microbiol). It would be interesting to investigate whether HEV entry requires cosignaling or coupling of EGFR with ITGB1 and we have added this aspect to the Discussion of the revised manuscript (lines 645-652).

In addition, we examined the effect of ITGB1 on EGFR expression by immunofluorescence staining in WT and ITGB1 KO cells and found that only one of the two ITGB1 KO clones had reduced EGFR expression (Revision Fig. 5). nHEV infection of both ITGB1 KO clones was reduced by 50% compared to WT cells (Fig. 1G), suggesting that reduced EGFR expression is not the main reason for the observed phenotype in these cells.

Revision Figure 5. Expression and cellular distribution of EGFR in S10-3 WT and ITGB1 KO cells. S10-3 cells were fixed and co-stained with antibodies against ITGB1 (magenta) and EGFR (green) and counterstained with DAPI (blue). The images were taken on a Zeiss Airyscan LSM900 confocal microscope and shown are maximum projections of 2 confocal slices (thickness = 0.25 μ m). Scale bar = 20 μ m.

18. An additional section should be added to discuss in more detail the biological relevance of ITGB1 during viral infection, how is the expression of this protein on hepatocytes related to the in vivo situation and to naked particles? knowing that particles transfer from gut to liver via the portal vein and are therefore enveloped? What is then the significance of this receptor?

HEV is transmitted via the faecal-oral route. It must therefore cross the intestinal barrier to reach its main site of replication, the liver. However, the form of HEV particles that reach the liver, i.e. naked or quasi-enveloped, remains to be determined. A study by Marion and colleagues (2020, Gut) suggests that HEV actively infects intestinal epithelial cells and that its progeny are released in their enveloped form. However, definitive in vivo evidence for HEV infection of the gut, e.g. in an in vivo model, is still lacking. Alternatively, small pathogens, including HEV, may reach the basolateral side of intestinal epithelial cells from the apical side via mechanisms such as transcytosis or through microbreaks in the barrier, potentially allowing naked particles access to the liver. In addition, ITGB1 is ubiquitously and abundantly expressed in various cell and tissue types, including intestinal cells (e.g. Jones et al.; 2006, JCB). As shown in new Suppl. Fig. 6B, we found that knockdown of ITGB1 in the intestinal cell line Caco-2 reduced nHEV infectivity but not eHEV infectivity (Suppl. Fig. 6C), suggesting that its role in HEV infection is not restricted to hepatocytes. Since the environment of the intestine favors the removal of the quasi-envelope, it is possible that HEV particles re-enters the body through naked particles released with the bile into the intestine. Therefore, the interaction between nHEV and ITGB1 may play a prominent role in HEV transmission or even persistence along the gut-liver axis. We have added these interesting points to the Discussion section of the revised manuscript (lines 638-644).

Minor points:

- Fig 1B and Fig 1F, why does the profile of ITGB1 protein differ between these two blots?

The two bands correspond to the two forms of ITGB1: the upper band at 130 kDa is the mature, fully glycosylated form of ITGB1 which localises to the plasma membrane, whereas the lower band at 110 kDa is the immature form found in the ER. Cell confluency has been shown to have a direct effect on the ratio between the two forms (Salicioni et al., 2004, JBC). High cell confluency appears to favor the accumulation of the 130 kDa form, as cell-cell contacts facilitate the transfer of $\beta 1$ integrin to the cell surface. Therefore, the different ITGB1 protein forms observed in our study were probably due to the fact that the lysates were harvested at different cell confluencies. We have repeated the Western blot analysis of WT and ITGB1 KO S10-3 cells to ensure that the ratio of the two bands is consistent throughout the manuscript and have updated Fig. 1F in the revised manuscript.

-Sup Fig1B, add intensity overlaps. Middle and low panels: scale bars are missing.

We have added intensity overlaps and scale bars to the panels in previous Suppl. Fig. 1B which corresponds to the new Suppl. Fig. 3B of the revised manuscript.

-Fig2B, please show in Sup Inf the images obtained with eHEV. Please also show images obtained with mock cells to assess the background of the green RNA probe.

We have supplemented the images in Fig. 2B with images obtained with eHEV particles (new Suppl. Fig. 13) as well as with mock-infected cells (Suppl. Fig. 7A).

-Fig2C, data should be presented as number of HEV particles/cell, as in Fig2B & Sup Fig5

We have updated this panel to show the number of HEV particles per cell in the revised Fig. 2.

-Fig 3E, please show data obtained with eHEV and KO cells.

We have performed colocalisation analysis between eHEV and ITGB1 (new Suppl. Fig. 9) as well as nHEV and ITGB1 in ITGB1 KO cells (Revision Fig. 6).

Revision Figure 6. Colocalisation of nHEV and ITGB1 in ITGB1 KO cells. Pre-chilled S10-3 ITGB1 KO cells were inoculated with nHEV (MOI = 20 GE/cells) and incubated on ice for 2 h on ice before fixation. nHEV particles were visualised by RNA-FISH (magenta) using the ORF1 probe followed by IF using an antibody against ITGB1 (green) and counterstain against DAPI (blue). Images were taken on a Zeiss Airyscan confocal microscope and shown are maximum projections of 2 confocal slices (thickness = 0.25 μ m). Scale bar = 10 μ m.

-Fig 6G: please add a legend. The involvement of Rab5c is suggested but could one elaborate on this in more detail.

Thank you for this suggestion. We have added a detailed legend to the new Fig. 6H (previous Fig. 6G) and mention the potential involvement of Rab5c in the entry process of nHEV particles (lines 1245-1251).

Reviewer #3 (Remarks to the Author):

The manuscript submitted by Fu et al., delves into an important step of the HEV life cycle, which I believe will be of interest to researchers working on HEV. nHEV, which is transmitted through feco-oral route, remains a major public health concern and identification of the host factor involved in virus entry is a significant step towards development of specific therapeutics. However, following concerns need to be addressed to improve the clarity and conclusions, as mentioned below.

We thank the reviewer's appreciation for our work, careful reading of the manuscript, and very useful and constructive comments.

1. Representative figures of FFU assays need to be shown to ascertain reliability of the assay.

Thank you for this important point. We have added representative images showing foci forming units of cells infected with nHEV in the presence or absence of BafA1 together with the corresponding percentage of infected cells (new Suppl. Fig. 2, please see also our response to Reviewer 1, # 2). In addition, we have included graphs showing all raw FFU counts per ml for all HEV infection assays in the new Suppl. Fig. 4 and refer to this throughout the manuscript.

2. The authors rely on FFU assay to interpret all the data on virus infection. In order to rule out the biasness of antibody towards nHEV, a western blot of ORF2 may be added in addition.

We appreciate the reviewer's insightful comment. To complement the FFU assays, we have performed Western blot analysis of lysates from drug and mock-treated nHEV-infected samples. As shown in the Revision Fig. 7, inhibition of endosomal acidification also reduced the detection of ORF2 in infected cells. Unfortunately, we were unable to perform the same assay for eHEV due to its low specific infectivity and therefore low levels of ORF2 expression in infected cells. However, we do not believe that the antibody has a bias against nHEV, as all FFU assays are based on the detection of replicating ORF2 and/or newly synthesised ORF2 within cells, regardless of the particle form used to infect the cells in the first place. These intracellular forms of ORF2 are not associated with the viral quasi-envelope as they are not viral particles.

Revision Figure 7: Endosomal inhibitors impair detection of ORF2 in nHEV-infected cells. S10-3 cells were pre-treated with the indicated concentrations of different endosomal acidification inhibitors 30 min prior to nHEV infection (MOI = 30 GE/cells). Virus inoculum was removed after 24 h and HEV infection was quantified by analysing cell lysates 5 days post-infection. Equal volumes of each sample were separated by SDS-PAGE and probed by Western Blot against ORF2 and β -actin.

3. With reference to line 349, “To ensure that viral particles packaged full-length HEV genomes”, it will be more convincing to incubate the cells for longer period, costain the cells with anti-ORF2 antibody and RNA-probe to demonstrate that functional genome was present. Partial genome containing ORF1 and ORF2 fragments cannot be ruled out by FISH.

We detected nHEV-infected cells 48 hours post-infection with probes targeting ORF1 and ORF2 RNA, and stained them with an antibody against the ORF2 protein (Revision Fig. 8). This analysis confirmed the co-detection of ORF1 RNA, ORF2 RNA, and the ORF2 protein within the same cell, verifying that our assay detects functional viral genomes. In addition, to ensure that the entry assay reflects productive infection, we have supplemented many of the results throughout the manuscript with FFU end-point infection data. However, as mentioned in our response (#14) to Reviewer 2, many but not all aspects can be analysed by analysing ORF2-positive events, as ORF2 abundance is highly dependent on successful HEV genome replication and potentially, accumulation of progenies.

Revision Figure 8. Viral inoculum used in the study contains functional HEV genomes. S10-3 cells were infected with nHEV (MOI = 20 GE/cell) for 48 h before fixation. ORF1 (yellow) and ORF2 (green) RNA particles were visualized by RNA-FISH (version 2 kit) followed by IF using an antibody against ORF2 (magenta) and counterstain using DAPI (blue). The images were taken on a Zeiss Airyscan LSM900 confocal microscope and shown are maximum projections of 2 confocal slices (thickness = 0.25 μ m). Scale bar = 10 μ m.

4. Refer to Fig 3H, the difference is not very striking. That should be explained.

We appreciate the reviewer for highlighting this important point. The mild phenotype observed in Fig. 3H may be attributed to two factors: First, to avoid severely impairing viral replication, we opted for a mild mutation (D to E) rather than a more disruptive alteration, such as mutating the D to an A (lines 455-456). As shown in our response to Reviewer 1 (# 4c), we used siRNA to knock down ITGB1 expression and conducted binding and infectivity assays using WT-nHEV and mutated D522E-nHEV (Suppl. Fig. 11A, B). Our results indicate that while the mutant virus was not entirely insensitive to ITGB1 depletion, its sensitivity was reduced compared to the WT virus. A similar trend was observed during D522E infection in the presence of the RGD peptide (Suppl. Fig. 11C, D), where the peptide exhibited a weaker inhibitory effect on the mutant virus compared to the WT. Taken together, these results suggested that the mild phenotype observed in Fig. 3H can be attributed to the subtle nature of the mutation, which does not induce a significant structural alteration in the capsid sufficient to completely abolish viral entry.

Second, as shown in Fig. 1G, knockout of ITGB1 results in only a ~50% reduction in nHEV infection, suggesting that the dependence of nHEV entry on the ITGB1 subunit is not absolute, which may explain why the mutant virus retains the ability to enter. Previous studies have shown that integrins can compensate for each other's functions, which may explain the relatively modest reduction in nHEV infection observed following ITGB1 knockout (Fig. 1G) and DGR mutation (Fig. 3G). Notably, some viruses use multiple pairs of integrins for entry (e.g. HCMV: Feire et al.; 2004, PNAS, or HSV: Gianni et al.; 2013, Plos Pathogens), allowing them to use alternative pathways in different cell types, thereby broadening their host range, as discussed in lines 620-631.

5. Refer to Fig 4A, it is expected to get empty particles, but why only RNA is visible (at least 2 dots)? That raises concern on technicality of the experiment.

The reviewer raises an important point. According to the manufacturer's protocol of the RNA-FISH assay used in our study, up to five dots per frame can be considered as background signals of the assay. If one zooms in carefully on Fig. 4A, only two smaller dots are visible, corresponding to the HEV RNA signal, which are not positive for the capsid signal. These dots are likely to fall into the category of non-specific background detection, especially since they only correspond to a tiny fraction of the total number of particles detected (see quantification panel in Fig. 4A). In our entry assay, because of the different background noise imposed by the cell and because we have much fewer signals per cell, we have optimised the signal-to-noise ratio to reduce this background to almost zero, as shown in the new Suppl. Fig. 7A.

6. Refer to Fig 4D, why BFA is not totally abrogating HEV RNA?

This observation is likely attributable to a mild contamination between fractions during the fractionation process, as illustrated in Fig. 4C. The imperfect separation of membrane and cytosolic fractions is a recognised limitation, which we now address more explicitly in the revised text (lines 540-541).

Also, BFA use need to be mentioned in the text while writing the result for 4D.

Thank you for pointing this out. As it fits better in the context of analysing the endocytic pathways used by HEV particles, we have moved the original Figs. 4C and D to the new Fig. 5 C,D of the revised manuscript and mention explicitly the use of BafA1 in the Result section (lines 537-540).

7. Refer to Fig 4, at 24 hour time point, progeny virus could also account for the signal.

To ensure that the viral genomes detected 24 h post-infection were from incoming viruses and not replicating genomes, we applied the nucleoside analogue NITD008 prior and for the duration of the entry assay. NITD008 has been shown to be an efficient replication inhibitor for several viruses, including HEV (Netzler et al., 2014, Antimicrob Agents Chemother; Jagst et al.; 2024, Antiviral Res). We have not detected any difference in the number of detected HEV RNA particles when analysing the NITD008-treated sample and the DMSO control, indicating that the detected genomes were not from progeny virus (new Suppl. Fig. 7E). We have added these results to the Results section of the revised manuscript (lines 413-416).

8. A recent study has shown that Rab11 positive endocytic recycling compartment serves as replication factory for HEV (doi: 10.1007/s00018-022-04646-y). If nHEV entry and assembly is dependent on Rab11, results need to be carefully interpreted. Authors need to explain their data accordingly.

Indeed, Rab11 has been shown to play a role in HEV replication and assembly (Bentaleb et al.; 2022, Cell Mol Life Sci) and for many viruses, entry and assembly pathways can be intricately related (e.g. beta coronaviruses study by Gosh et al.; 2020, Cell). For these reasons, we believe that studies of the role of Rabs, for example by downregulation, based on full HEV infection assays are really difficult to interpret (please see our responses to Reviewer 1, #3 and 2, #14, Revision Figure 4). This altogether highlights the need to use an assay that clearly separates the cell entry steps from the later stages of the viral life cycle. Throughout our study, we examined HEV particles by detecting RNA and/or capsid within a timeframe of 8-24 hours post infection. Thanks to this Reviewer's suggestion, we have used a replication inhibitor (as shown in response to the comment above, new Suppl. Fig. 7E) to prove that within this timeframe, we only study the early steps of the HEV life cycle. Therefore, this approach ensures that the involvement of cellular factors such as Rab proteins and integrins is specifically linked to HEV entry, rather than subsequent stages of the viral life cycle, such as replication and/or assembly of progeny.

We fully agree with the Reviewer that this interesting point has to be explained and we have added it to the Discussion section of the revised manuscript (lines 694-705).

A replication defective mutant virus may help in distinguishing between entry and progeny virus assembly steps.

We appreciate this valuable suggestion. Unfortunately, however, attempts to package replication-defective genomes into infectious particles by trans-complementation have been unsuccessful due to the lack of replication and therefore of sufficient viral genomes.

9. Refer to Fig 4H, why is there a break in x axis? It seems not to affect the representation.

We introduced the break solely for aesthetic reasons and space constraints within the previous Fig. 4. However, meanwhile, we have decided to remove this panel and to replace it with the suggested experiment in the next comment.

10. Refer to Fig 4H, is it possible to supplement the data with HEV RNA RT-PCR of membrane fraction?

Thank you for your suggestion. During the revision process, we have removed Fig. 4H (see response to Reviewer 2, #3) and replaced it with a membrane fractionation experiment to analyse the partitioning of HEV genomes over time (3h vs 24h post-binding, see new Figs. 4C & 4D, lines 509-513). This new analysis shows that the majority of HEV genomes have entered the cytoplasm only after 3 h post-infection. Unfortunately, we were unable to perform the same experiment for eHEV due to the low titers. We have moved the previous Figs. 4C and 4D to the new Fig. 5C and 5D in the revised manuscript.

Point-by-point reply to the reviewers' comments

Reviewer #1:

The revised manuscript has been greatly improved in the clarity and completeness of several experiments. However, several issues remain to be addressed.

We are grateful for the appreciation of our efforts made during the previous revision. We have addressed the remaining issues below.

1. A major concern is that knockout of ITGB1 led to only a 2-fold reduction in nHEV cell binding and infectivity. Thus, ITGB1 doesn't appear to play a critical role in nHEV entry. This rather moderate effect diminishes the significance of the study.

Reviewer #1 has previously seen the 2-fold reduction in nHEV binding and infection in the absence of ITGB1. We have previously acknowledged the mild phenotype in ITGB1 KO cells (line 630) we have likewise previously discussed that in the absence of ITGB1, other beta integrins are being upregulated to compensate the loss of ITGB1 (lines 626-630). Critically, ITGB3 and ITGB5 heterodimers, similar to ITGB1 heterodimers, bind to their ligands via their RGD motif. We have added this additional information together with the respective citations to the revised manuscript (lines 630-631).

2. The direct binding between ITGB1 and the viral capsid remains unconvincing. New data are provided to support direct binding using Alphafold modeling. However, the model based on ORF2 dimer indicates the ITGB binding site on the virion is inaccessible and therefore conformational changes in the virion are likely needed. What could be the trigger for such conformational changes? The authors should use HEV-LP to demonstrate direct binding between the capsid and ITGB1 and the role of the DGR motif using various DGR mutants.

The need for a direct interaction would imply that ITGB1 acts as a cell entry receptor for nHEV particles, a claim we were careful not to make in our manuscript. Yet, we believe we have already provided very strong evidence for this interaction. To address this concern, which was raised also by Reviewer #2 initially, we designed and carried out a proximity ligation assay (PLA) in order to determine whether purified, authentic, infectious nHEV particles bind in close proximity to ITGB1 expressed on the cell surface (Fig. 3J). This assay has been performed in numerous studies to demonstrate protein-protein interactions (for example, but not limited to: Hervouet *et al.*; 2022, PLoS Pathog; Kurdi *et al.*; 2016, Nature Commun; Su *et al.*; 2022, J Biomed Sci; Söderberg *et al.*, 2006, NatMethods; Halder *et al.*; 2020, Nat Commun). The quantitative analysis of our PLA data, as shown in the righthand panel of Fig. 3J, provides very strong evidence that the ORF2 capsid protein that is present within authentic WT nHEV particles binds to the cell surface in very close proximity/or in direct interaction with ITGB1. By contrast, the proximity of the mutant D522E capsid to ITGB1 was reduced from that observed with the WT infectious particle. In order to provide additional experimental evidence of the close proximity/interaction on the cell surface, we have performed a quantitative colocalisation analysis in this second revision round to further support the specificity of the interaction between WT nHEV particles and ITGB1. As shown in new Suppl. Fig 9A, B, the colocalisation of D522E nHEV particles with ITGB1 was reduced as compared to WT nHEV particles (Fig. 3E), therefore confirming our previous results with the PLA assay.

To develop an alternative approach to demonstrate this interaction using HEV-LPs as a surrogate for the WT native virus particle makes little sense to us as it would not provide data that are any more conclusive than the PLA data we have present. HEV-LPs are assembled from N- and C-terminally truncated ORF2 capsid proteins (~500 amino acids) and they are thus significantly different from and less authentic than the genuine infectious virus particles (assembled by the full-length ORF2 protein of 660 amino acids) we used in the PLA. HEV-LPs assemble into a smaller 60-mer capsid compared to the 180-mer native capsid and therefore may not fully replicate the characteristics of infectious virions (reviewed in

Brüggemann *et al.*; 2025, Trends Microbiol). Therefore, we believe it would neither change nor significantly strengthen the conclusions we reach in the paper.

We understand that the AlphaFold model may have appeared confusing. However, the proposed ITGB1 binding site could actually be well accessible due to a flexible, proline-rich hinge region between the P- and M-domain (see Revision Figure), as suggested by the cryo-EM analysis of the HEV-LP structure (Guu *et al.*; 2019, PNAS). We have clarified this in the revised manuscript, together with explicitly stating that we only used the head domain of ITGB1 to model the interaction (lines 475 and 482-485). We have added a statement to the revised manuscript that more work needs to be done to make further conclusions about the predicted interaction model (lines 494-495).

Revision Figure: 3D structure of the capsid protein (HEV3 PDB ID 2ZTN), adapted from Cancela *et al.*; 2022, Arch Virol.

3. It remains unclear how nHEV moves from the recycling endosomes to the lysosome for uncoating. Additional experimental evidence would be needed, as this seems to be an important aspect of the working model shown in Fig. 7. Also, the role of Rab5c in nHEV entry should be addressed to strengthen the model.

We agree with the Reviewer that this is an aspect of our study that we have not yet addressed experimentally. Numerous receptors and viruses enter cells via the recycling endosome and subsequently traffic through late endosomes and/or lysosomes. We have already discussed in the previous version of the manuscript (lines 689-695) that a possible link between the recycling endosome and the endolysosome is the process of autophagosome formation. Rab11 is thought to regulate the fusion of MVBs with autophagosomes, and LAMP1 is known to be present in autophagic organelles. Additionally, Rab7 has been demonstrated to be important for the autophagic pathway and the maturation of autophagosomes. It is possible that the autophagic machinery could promote the recruitment of molecular motors to the ruptured endosome and thereby assist in the exit of the HEV genome from these compartments. We have added in the revised manuscript that future studies should clarify the role of autophagosomes in HEV cell entry (lines 695-696). We believe that the assay we have established in this work allows such studies to be performed. However, addressing this aspect experimentally at this stage will significantly delay the publication of this assay, which we believe will be very useful to the HEV community.

In addition, we have removed Rab5c from the working model and also explicitly mention in the Discussion section that the potential involvement of Rab5c should be addressed experimentally in the future (lines 683-684).

Reviewer #2:

Compared to the previous version, Fu and colleagues have significantly improved their manuscript. They have carefully and convincingly addressed my concerns, and implemented corresponding changes in the manuscript. Therefore, I believe that their manuscript is suitable for publication in Nature Communications.

We thank the reviewer for their constructive feedback and positive evaluation of our study. We are also grateful for their careful reading and constructive suggestions. We have incorporated them in the revised manuscript.

Reviewer #3:

The revised manuscript addresses all my queries and I am satisfied with the response of the authors. I recommend publication of this manuscript.

We thank Reviewer #3 for their help in improving the manuscript and their recommendation for publication.